# MotifScreen: Generalizing Virtual Screening through Learning Protein-Ligand Interaction Principles

## Abstract

Virtual screening methods continue to face a fundamental trade-off between accuracy and efficiency. Deep learning-based methods attempting to address this challenge suffer from overfitting due to sparse and biased training data and inadequate validation practices. We first show that the over-optimism in prevalent deep learning-based methods is due to incorrect validation setups, and their actual performance approaches that of random selection. We then present MotifScreen, a structure-based end-to-end virtual screening method that addresses these limitations through principle-guided multi-task learning. We ask our network to rationalize the prediction by understanding the principles of protein-ligand interactions in a step-by-step manner: 1) receptor pocket analyses, 2) ligand-pocket chemical compatibility, and 3) ligand binding probability given its compatibility. This multi-task framework, trained on a new dataset specifically curated for the task, significantly outperforms existing methods and classification-only baselines when evaluated on a stand-alone test set.

## 1 Introduction

Deep learning (DL) methods for structure-based virtual screening (SBVS) face a credibility crisis. Despite the increasing sophistication of model architectures, their reported performance on academic benchmarks has shown a counterintuitive pattern. While recent, advanced models often report an AUROC around 0.8 (Zhang et al., 2023; Moon et al., 2022; Cao et al., 2025), some preliminary methods published years prior claimed near-perfect scores exceeding 0.9 (Chen et al., 2019). This inconsistency raises serious questions about the validity of these metrics and the evaluation practices used in the field.

This validation problem comes at a pivotal moment. The convergence of accurate, large-scale protein structure prediction, exemplified by AlphaFold (Abramson et al., 2024), and the availability of massive make-on-demand chemical libraries like Enamine REAL Space (Grygorenko et al., 2020) has expanded the searchable chemical space by billions of compounds. This new landscape presents an unprecedented opportunity for discovering novel therapeutics. However, the sheer scale of these libraries makes traditional physics-based docking methods such as AutoDock (Eberhardt et al., 2021) prohibitively slow. For this reason, establishing reliable evaluation methods for newer, faster screening models is essential for setting a clear standard for future research and ensuring these tools are genuinely useful.

The crisis stems from the outdated benchmarks used for model validation. Popular datasets such as DUD-E(Mysinger et al., 2012), DEKOIS 2.0(Bauer et al., 2013) and CASF-2016(Su et al., 2018) were designed before the widespread adoption of deep learning models. These benchmarks represent a small and often-biased fraction of the vast and diverse chemical space now available for screening, and they lack robust safeguards against common DL failure modes like data memorization and shortcut learning. Numerous studies have now demonstrated that DL models can achieve top-tier performance on these benchmarks by exploiting subtle biases and memorizing ligand features, often without learning the fundamental principles of protein-ligand binding (Wallach & Heifets, 2018; Chen et al., 2019; Sieg et al., 2019). A model that cannot generalize beyond a flawed benchmark is of little use for real-world campaigns that require screening vast and diverse chemical spaces. Addressing

this limitation requires both identifying the sources of bias and developing evaluation-aware methods that resist these shortcuts.

In this work, we address this challenge on two fronts. First, we introduce **ChEMBL-LR**, a new benchmark carefully curated to remove identified biases. Second, we propose **MotifScreen**, a new SBVS method designed to resist learning such shortcuts. MotifScreen is a structure-based, end-to-end framework designed for robust generalization. Unlike models that perform simple classification, MotifScreen is trained to reason about protein–ligand interactions through a principle-guided multi-task pipeline: 1) identifying pocket motifs, 2) assessing ligand compatibility with those motifs, and 3) predicting the final binding probability. This approach is enabled by a carefully curated training dataset, discouraging ligand memorization and mitigating decoy bias. Validation of such approach was done using our leakage-resistant benchmark.

- **Critical Analysis.** We provide evidence for the over-optimism of current methods by analyzing data leakage and its impact on the performance of existing deep learning VS models.
- **Leakage-Resistant Dataset.** We develop **ChEMBL-LR**, a novel benchmark dataset for more realistic and reliable model evaluation.
- **Robust Architecture.** We propose **MotifScreen**, a principle-based multi-task learning model that leverages multi-modal data to achieve generalization for virtual screening.
- **Evaluation.** We present a performance comparison demonstrating that MotifScreen is competitive with state-of-the-art models on traditional benchmarks while showing significantly better robustness and generalization on our new ChEMBL-LR dataset.

## 2 RELATED WORKS

### 2.1 DEEP LEARNING APPROACHES FOR STRUCTURE-BASED VIRTUAL SCREENING

Structure-based virtual screening (SBVS) uses 3D structural information to predict protein-ligand interactions, offering a powerful alternative to ligand-based methods. Traditional docking have been foundational to SBVS, but remain slow and moderately accurate, limiting their scalability to modern ultra-large libraries. Deep learning-based SBVS models offer a faster alternative, typically falling into two main categories.

**Scoring and Docking Models.** Models such as KarmaDock (Zhang et al., 2023), SurfDock (Cao et al., 2025), and TANKBind (Lu et al., 2022) predict both binding pose and affinity, often incorporating geometric inductive biases like triangle attention following works of Jumper et al. (2021). Scoring-focused models such as AK-Score2 (Hong et al., 2024) and PIGNet (Moon et al., 2022) take pre-docked complexes as input, but dependent on external docking tools, inheriting their inherent limitations.

**Protein Complex Structure Prediction Models.** Inspired by breakthroughs in protein structure prediction such as AlphaFold2 and 3 (Jumper et al., 2021; Abramson et al., 2024), models like (Passaro et al., 2025) aim to predict binding affinity as well as the entire complex structure *de novo*. Despite their accuracy in structure prediction, they remain too computationally intensive for large-scale virtual screening applications.

This leaves a critical gap in the field: the need for a method that is both efficient and can learn from a wider range of biochemical data beyond the limited pool of experimental structures. Our proposed method, **MotifScreen**, is a multi-task framework designed to address this gap by learning from both structural and non-structural activity data to achieve better generalization.

### 2.2 THE CHALLENGE OF UNBIASED EVALUATION

Despite the promising reported scores, real-world utility of DL models is often questionable. Their success is frequently measured on flawed benchmarks that suffer from systemic biases, leading to over-optimistic and misleading results. This bias problem is pervasive across widely-used benchmarks, creating systematic evaluation flaws. Two problems are particularly damaging to the field: **data leakage** and **ligand bias**.

**Data Leakage.**     Benchmarks such as DUD-E (Mysinger et al., 2012) and CASF-2016 (Su et al., 2018) share many of the same protein targets and active ligands found in the general training sets like PDBbind (Liu et al., 2017) (Section 4.1 and Appendix A.1). This overlap enables models to achieve inflated scores by memorizing ligands instead of learning interaction principles.

**Ligand Bias.**     *Bias in actives* arises from selecting actives with much higher affinities than typical virtual screening hits, making benchmarking less transferable to real practice. *Bias in decoys* is also problematic. Most benchmarks generate decoys by selecting molecules with similar physicochemical properties (e.g., molecular weight, logP) but different topology from the known actives.  While well-intentioned, this creates a subtle but powerful bias: the active compounds for a given target often share similar core scaffolds, while the decoys are, by design, topologically dissimilar.

This makes the classification task artificially easy.  Instead of learning to recognize complex 3D complementarity between a ligand and a protein pocket, a model can learn a simple shortcut: distinguishing between the "active-like" and the "decoy-like" clusters in the chemical space.  The model is rewarded for being a good ligand feature detector, not a good protein-ligand interaction predictor.

**Asymmetric Validation Embedding (AVE).**     Introduced by Wallach & Heifets (2018), Asymmetric Validation Embedding (AVE) bias measures the statistical redundancy between training and test set ligands. AVE bias considers similarity across both actives and inactives $(A_T A_V - I_T A_V) + (I_T I_V - A_T I_V)$. A large $(A_T A_V - I_T A_V)$ term indicates strong active bias, while a large $(I_T I_V - A_T I_V)$ term indicates strong decoy bias. A high overall AVE bias suggests a benchmark is rewarding ligand memorization, not generalization.

### 2.3   EFFORTS TO ADDRESS EVALUATION BIAS

In response to these challenges, researchers have developed more rigorous benchmarks. A notable example is LIT-PCBA, which was designed to be a less-biased dataset for machine learning models (Tran-Nguyen et al., 2020).  It uses the AVE procedure to create unbiased internal training and validation splits.

However, LIT-PCBA has key limitations for evaluating true generalization to novel systems. First, its scope is narrow, covering only 15 protein targets. Second, its unbiased nature is compromised when models are trained on common external datasets like PDBbind. Using an external training set breaks the original AVE-optimized split and invalidates the benchmark's statistical assurances. This is particularly problematic because all 15 targets are also present in large pre-training datasets, creating a high risk of target-based data leakage.

Therefore, a need remains for a benchmark that not only controls for internal ligand bias but also prevents target-based leakage by enforcing a strict separation of protein targets from common training datasets.  Our new leakage resistant dataset, ChEMBL-LR is designed and tested to meet such a need. ChEMBL-LR addresses both issues through strict target-wise separation and property-matched cross-decoys, providing a rigorous evaluation protocol for DL-based VS methods.

## 3   METHODS

Our work addresses evaluation bias through two complementary strategies: 1) a multi-task architecture **MotifScreen** that learns generalizable interaction principles rather than dataset-specific shortcuts, and 2) a rigorously curated benchmark **ChEMBL-LR** that prevents the biases identified in existing evaluations.  This section details the design philosophy of our model and the construction of our dataset.

### 3.1   MOTIFSCREEN – MODEL ARCHITECTURE

The core design philosophy of MotifScreen is to move beyond simple binary classification and resist the evaluation biases. Rather than learning dataset-specific shortcuts, the model is forced to understand protein-ligand interactions through three complementary reasoning steps: identifying chemical interaction motifs in binding pockets, assessing structural compatibility between ligands

and pockets, and predicting binding probability based on this compatibility. This multi-task design prevents the model from relying on simple ligand memorization or topological clustering.

MotifScreen (Figure 1) processes protein and ligand graphs through parallel geometric encoders that generate SE(3)-equivariant representations. The protein binding pocket is represented using virtual grid points that capture the spatial organization of potential interaction sites. The multi-task framework then operates on these representations to learn meaningful protein-ligand interactions, with detailed architectures provided in Appendix C.

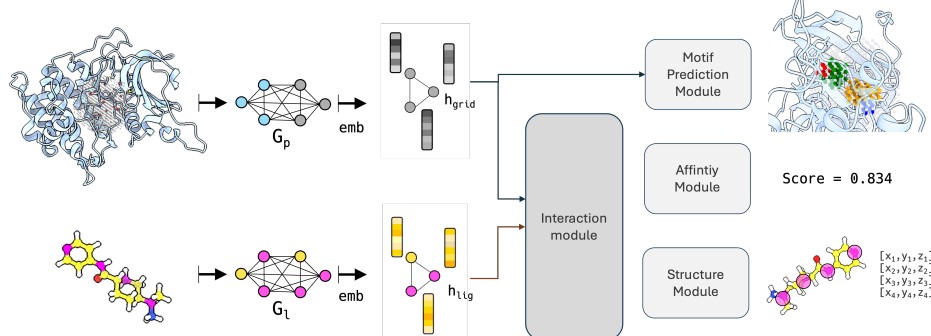

Figure 1: **Architecture of MotifScreen.** MotifScreen generates two key representations; a receptor embedding that captures motif positions and binding site chemistry, and an interaction embedding that encodes protein-ligand structure and affinity information. The representations are used to predict motifs, ligand coordinates and binding scores for screening. Detailed architecture of the interaction module is in Figure C1.

### 3.2 MULTI-TASK LEARNING FRAMEWORK

The multi-task learning framework improves generalization through complementary learning objectives. Each task addresses a specific aspect of protein-ligand interaction: structure prediction learns physically plausible ligand and protein-ligand complex geometry, motif prediction identifies chemical interaction patterns in protein pockets, and binding affinity prediction integrates both chemical and spatial complementarity. This design makes it difficult for the model to exploit dataset shortcuts while encouraging physically meaningful representations that transfer across different targets and chemical spaces.

The total training loss is a weighted sum of objectives from three primary tasks, implemented through dedicated modules in an end-to-end fashion. Importantly, the structure and motif prediction tasks are only evaluated on training data with available bound structures, while the affinity module is trained on all data, including activity-only samples without structural information.

**Motif Classification – Motif Module.**    This module predicts chemical interaction patterns (motifs) at binding surface grid points, and is responsible for interpretation of receptor pocket chemistry. The module outputs marginal probability of six binding motifs: H-bond donor, acceptor, donor-and-acceptor, aromatic, aliphatic and none.

This module is based on MotifGen (Anonymous, 2025) which predicts motif using protein-protein interaction data. In MotifScreen, probability is inferred across all grid points at once to enable end-to-end learning, as opposed to the prior network where inference was done per grid at a time. Motif probability is predicted from the grid embedding calculated by an SE(3) equivariant transformer (Fuchs et al., 2020) which updates protein grid features from neighboring residue atoms. This task regularizes the model, compelling the receptor embedding to learn a meaningful, high-level representation of the binding pocket. The main training loss is a masked binary cross-entropy loss applied to the predicted motif representations of the protein surface.

**Key Atom Positioning – Structure Module.**    This module (Algorithm 5) predicts the 3D positioning of key ligand atoms representing molecular fragments, and is responsible for interpretation of chemical compatibility between the pocket and ligand. This fragment-level representation has two advantages. First, in contrast to structure-agnostic methods, the structural information can be

effectively incorporated into ligand binding evaluation. Second, it tolerates small structural variations (e.g., side-chain movements) that might cause sensitivity issues in all-atom docking methods. Details on key atom selection are in Appendix C.2.1.

The model is trained with losses that enforce physically plausible molecular geometry, including distance between predicted and ground-truth key atom positions and a pair-distance loss that penalizes incorrect intra-ligand distances.

**Binding Affinity Scoring – Affinity Module.** This module predicts a binding score, which is the ultimate goal of a virtual screening model. The module takes the pair embedding which is also used for structure module, and a global ligand embedding that captures properties like molecular size. Training uses three loss types: classification loss, ranking loss, and contrastive loss to optimize discrimination between active and inactive molecules.

Detailed explanations of each module are provided in Appendix C with loss functions in Appendix C.5, including description of the training scheme in Appendix C.6.

### 3.3 Leakage Resistant Dataset Construction

To address the data leakage and decoy bias problems identified in Section 2.2, we developed **ChEMBL-LR** through a curation protocol that enforces strict separation between training and evaluation data while controlling for chemical bias.

Our curation addresses four key challenges: (1) **Target-wise separation** prevents protein family memorization by ensuring no homologous proteins appear in both training and test sets, (2) **Realistic active selection** ensures affinity distribution of actives mirror real-case VS scenarios, (3) **Unbiased decoy selection** uses cross-decoys (actives for other targets) while chemical space balance is verified through AVE bias analysis, and (4) **Removal of non-drug-like molecules** removes outliers like glycans that create unwanted chemical biases. These strategies are implemented through systematic data processing detailed in Appendix B.1.2. The resulting ChEMBL-LR benchmark contains 60 targets, achieving a mean AVE bias of 0.033 compared to 0.17 for DUD-E, with no target leakage.

**Training Data Sources.** We combined structural and activity data from multiple sources: PDBbind (Liu et al., 2017) and BioLip (Zhang et al., 2024) for structural complex with activity data, and ChEMBL 34 (Gaulton et al., 2012) for additional activity data, adding over 40,000 protein-ligand pairs. We augmented the structural data using a "cross-docking" strategy where actives for homologous receptors (sequence identity > 0.95) serve as putative actives for each other, providing consistent chemical signals across highly similar proteins (Martin, 2010).

**ChEMBL-LR Benchmark.** The benchmark was curated from a completely separate partition of ChEMBL with no homologous proteins in the training data. This strict separation ensures unbiased evaluation of generalization to novel targets. A comprehensive description of all data processing, filtering, and splitting procedures is provided in Appendix B.1.

## 4 Experiments

In this section, we first demonstrate the need for a leakage-resistant benchmark and present our solution. We then compare MotifScreen's performance on this new benchmark against traditional methods, linking its design to its generalizability. Finally, we validate its core architectural philosophy through ablation studies.

### 4.1 Benchmark Analysis and ChEMBL-LR Development

#### 4.1.1 Systematic Biases in Existing Benchmarks

Deep learning-based VS methods achieve inflated performance due to systematic biases in popular benchmarks. To create a fair evaluation standard, we first analyzed the systemic biases in popular benchmarks and then developed ChEMBL-LR, a new benchmark designed to overcome the two main flaws we identified.

**Target leakage.** DUD-E (94%), DEKOIS 2.0 (98%), and even the modern LIT-PCBA (100%) have similar proteins (sequence identity > 0.4) in PDBbind (Liu et al., 2017), the primary training source for most DL methods (Table A2). This allows models to succeed by recognizing familiar targets rather than by generalizing binding principles.

**Ligand bias.** Actives and decoys create easily separable clusters in chemical space that reward ligand memorization over protein-ligand interaction learning. We quantified ligand bias using the Asymmetric Validation Embedding (AVE) analysis. As shown in Table 1, older benchmarks like DUD-E show a high AVE bias of 0.17, indicating that models can distinguish actives from decoys based on simple ligand features alone. While LIT-PCBA shows low AVE bias, this was calculated with their internal training-validation split.

### 4.1.2 ChEMBL-LR: A Leakage-Resistant Benchmark by Design

Our ChEMBL-LR benchmark was constructed to directly address these issues. By design, no protein target in ChEMBL-LR has a homologous protein in our training set (PDBbind, BioLip, non-benchmark ChEMBL targets) eliminating the possibility of target leakage. Furthermore, our curation protocol ensures that actives and decoys are homogeneously distributed. AVE analysis confirms this, showing a near-zero mean bias of 0.033 (Table 1), which indicates that simple ligand memorization is not a viable shortcut. To further validate that ChEMBL-LR is genuinely challenging, we conducted two key experiments.

**Ligand Bias Resistance.** To validate that ChEMBL-LR resists ligand memorization, we trained several ligand-only ML models (kNN, Random Forest, Logistic Regression) using ECFP4 fingerprints on our training set. On ChEMBL-LR, these models achieved a near-random mean AUROC of $0.530 \pm 0.156$, with Logistic Regression resulting in mean AUROC of 0.509. This confirms that ChEMBL-LR cannot be solved through simple ligand memorization, requiring genuine understanding of protein-ligand interactions.

**Data Leakage Resistance.** To validate that ChEMBL-LR resists protein, ligand, and protein–ligand data leakage, we trained a Random Forest model using both protein (ESM2 embeddings (Lin et al., 2023)) and ligand (ECFP4 fingerprints) features. To mirror the training setup of popular DL methods, the model was trained on PDBbind. We then evaluated this model across four benchmarks: DUD-E (Mysinger et al., 2012), DEKOIS 2.0 (Bauer et al., 2013), LIT-PCBA (Tran-Nguyen et al., 2020), and ChEMBL-LR(ours).

Table 1 reveals a clear trend. On DUD-E and DEKOIS 2.0, which suffer from both high ligand bias and target leakage, the protein-aware RF model achieves an inflated AUROC of 0.691 and 0.657, respectively. On LIT-PCBA, which controls for ligand bias but still contains significant target leakage, the task becomes harder and the model's performance drops to 0.542. Finally, on ChEMBL-LR, which controls for both bias types, the model's performance collapses to $0.518 \pm 0.131$, a result statistically indistinguishable from random. This provides strong evidence that ChEMBL-LR successfully measures generalization to novel targets rather than rewarding dataset memorization.

Table 1: AVE bias comparison across benchmarks. Lower absolute values indicate reduced ligand memorization and more rigorous evaluation.

| Dataset | AVE bias ($\downarrow$) | RF AUROC ($\downarrow$) | # Targets |
|---|---|---|---|
| DUD-E (2012)[a] | 0.17 | $0.691 \pm 0.170$ | 102 |
| DEKOIS 2.0 (2013)[a] | 0.11 | $0.657 \pm 0.198$ | 80 |
| LIT-PCBA (2020)[b] | **0.016** | $\underline{0.542 \pm 0.065}$ | 15 |
| **ChEMBL-LR (Ours)** | $\underline{0.033 \pm 0.192}$ | **0.518 ± 0.131** | 60 |

[a] Median AVE calculated following the evaluation protocol by Imrie et al. (2021) (per-target CV on dataset-specific "unbiased" properties). Adopted from the cited work.
[b] Mean AVE bias adopted from (Tran-Nguyen et al., 2020).

## 4.2 VIRTUAL SCREENING BENCHMARKS

We evaluate MotifScreen against existing methods on both **ChEMBL-LR** and DUD-E benchmarks. To ensure a truly fair evaluation, we removed all training data with high similarity to DUD-E targets (sequence identity $> 0.4$ and ligand Tanimoto similarity $> 0.4$, Table B2). This prevents MotifScreen from benefiting from the target leakage that affects many existing models as we criticized. In all the evaluations, the sigmoid-transformed value of the MotifScreen scalar output $\hat{y}$ was used as binding score. Training algorithm and details of MotifScreen used for this benchmark are in Appendix C.6.

Table 2: Performance Comparison on the ChEMBL-LR Benchmark.

| Model | AUROC | BEDROC ($\alpha = 20$) | EF1% |
|---|---|---|---|
| AutoDock-Vina | $0.541 \pm 0.125$ | $0.075 \pm 0.075$ | $2.189 \pm 3.439$ |
| AK-Score2 | $0.527 \pm 0.135$ | $0.054 \pm 0.069$ | $0.803 \pm 2.308$ |
| KarmaDock | $0.527 \pm 0.135$ | $0.042 \pm 0.047$ | $1.317 \pm 2.715$ |
| SurfDock | $0.576 \pm 0.151$ | $0.090 \pm 0.097$ | $3.443 \pm 6.076$ |
| RF (ESM+FP) | $0.518 \pm 0.131$ | $0.093 \pm 0.089$ | $2.09 \pm 3.20$ |
| **MotifScreen** | $\mathbf{0.680 \pm 0.165}$ | $\mathbf{0.146 \pm 0.205}$ | $\mathbf{4.16 \pm 5.65}$ |

Table 3: AUROC and normalized enrichment factor (NEF) across benchmarks.

| Model | Ref. Set | AUROC | | EF1% (Normalized) | | | Robustness |
|---|---|---|---|---|---|---|---|
| | | Ref. | ChEMBL | Ref. Raw | Ref. NEF[a] | ChEMBL NEF[b] | NEF Ret.[c] |
| AutoDock-Vina | DUD-E | 0.720 | 0.541 | 9.70 | 0.46 | 0.07 | 15.3% |
| AK-Score2 | DUD-E | – | 0.527 | 14.60 | 0.70 | 0.03 | 3.7% |
| KarmaDock | DUD-E | 0.754 | 0.512 | 15.87 | 0.76 | 0.04 | 5.6% |
| SurfDock | DEKOIS | 0.758 | 0.576 | 18.17 | 0.59 | 0.11 | 19.0% |
| RF (ESM+FP) | DUD-E | 0.691 | 0.518 | 14.21 | 0.68 | 0.07 | 10.0% |
| **MotifScreen** | DUD-E | 0.753 | **0.680** | 5.94 | 0.28 | **0.13** | **47.4%** |

**Note:** ChEMBL-LR results are from our experiments; Reference (Ref.) results are from cited publications, details in Table D5.
[a] NEF = Raw EF/21.0 (for DUD-E) or /31.0 (for DEKOIS).
[b] NEF = Raw EF/31.0 (for ChEMBL-LR).
[c] NEF Retention = $\text{NEF}_{\text{ChEMBL}}/\text{NEF}_{\text{Ref}} \times 100$.

### 4.2.1 PERFORMANCE ON CHEMBL-LR

Table 2 and 3 compares MotifScreen with physics-based docking (AutoDock-Vina (Eberhardt et al., 2021)), DL-based docking (KarmaDock (Zhang et al., 2023), SurfDock (Cao et al., 2025)), DL scoring models (AK-Score2 (Hong et al., 2024)), and an RF baseline (details in Appendix A.2).

MotifScreen achieves a mean AUROC of 0.680, an 18% relative improvement over the strongest baseline, SurfDock (0.576; Cohen's $d = 0.53$). Most baselines perform near-random (AUROC $\sim$0.51–0.54), making our gains substantial (AUROC effect sizes $d = 0.60$–0.97). Improvements are statistically significant for AUROC across all baselines (Wilcoxon signed-rank, BH-corrected $p < 10^{-4}$; Appendix Table D3) and for EF1% against most methods ($p < 0.05$ vs AutoDock-Vina, KarmaDock, AK-Score2, RF). In terms of BEDROC ($\alpha = 20$), which evaluates early enrichment, MotifScreen (0.146) outperforms SurfDock (0.090) and AutoDock-Vina (0.075).

While MotifScreen shows substantial improvements in overall discrimination (AUROC), the early enrichment results (EF1%) are more nuanced. The difference in EF1% versus SurfDock is not statistically significant ($p = 0.16$). However, MotifScreen demonstrates promising consistency across targets: ranking first on 21/60 targets (35%) for EF1% compared to SurfDock's 11/60 (18%), and achieving the highest AUROC on 31/60 targets (52%) versus SurfDock's 8/60 (13%) (Figure D1).

MotifScreen is also highly efficient: $\sim$0.03s per compound on four GPUs, versus $\sim$10s for SurfDock, making it practical for ultra-large library screening (Appendix E.1).

### 4.2.2 Performance on External Benchmarks

On DUD-E, MotifScreen achieves AUROC 0.753 and retains 90% of this performance on ChEMBL-LR (0.680). While baselines like KarmaDock drop to near-random AUROC ($\approx 0.5$) on ChEMBL-LR, MotifScreen retains significant predictive power. (Table 3). We also compared models' enrichment factors between the two benchmarks. EF1% values across benchmarks are inherently biased due to differing active-to-decoy ratios, which dictate the theoretical maximum EF ($EF_{max} = 1/\text{Active Ratio}$). Specifically, DUD-E (1:20 ratio) has an $EF_{max}$ of 21.0, while ChEMBL-LR and DEKOIS 2.0 (1:30 ratio) have an $EF_{max}$ of 31.0. To address this, we converted raw scores to the Normalized Enrichment Factor (NEF), defined as $NEF = \text{Raw } EF/EF_{max}$ (Liu et al., 2018). Baseline methods exhibit a severe collapse in NEF when applied to ChEMBL-LR. For instance, KarmaDock achieves a high NEF of 0.76 on DUD-E but drops to 0.04 on ChEMBL-LR, retaining only 5.6% of its normalized performance. In contrast, MotifScreen demonstrates the highest robustness, retaining 47.4% of its NEF, significantly outperforming all baselines in generalization capability. The smaller degradation in performance for MotifScreen indicates stronger generalization to novel targets and chemical spaces, rather than reliance on dataset-specific shortcuts.

Detailed performance statistics for MotifScreen on both ChEMBL-LR and DUD-E benchmarks are provided in Appendix Tables D1 and D2. For a comprehensive comparison against other virtual screening methods, Table D5 presents the absolute AUROC and EF1% scores across the ChEMBL-LR, DUD-E, and DEKOIS 2.0 benchmarks.

**Note on Experimental Fairness.** While most baseline methods benefit from target overlap with DUD-E (as shown in Section 4.1), MotifScreen deliberately excludes such targets from training dataset. This conservative approach likely underestimates MotifScreen's relative advantage.

**Note on Training Data Modalities.** The disparity in training regimes between MotifScreen and the baselines stems from a fundamental architectural difference. Structure-based baselines (e.g., SurfDock, KarmaDock) strictly require ground-truth 3D complex structures, limiting their training to structural datasets like PDBbind where leakage is difficult to fully eliminate without harming model performance. In contrast, MotifScreen is designed to learn from bioactivity datasets (ChEMBL) lacking experimental structures in addition to structural datasets. This capability allows us to construct a strictly leakage-free training set with a larger amount of protein-ligand interaction data that structure-dependent baselines physically cannot utilize. Therefore, this comparison highlights MotifScreen's unique architectural advantage: the flexibility to leverage broader, unbiased chemical spaces for robust generalization.

### 4.3 Ablation Studies and Model Analysis

We conducted ablation studies to validate MotifScreen's core architectural design and understand how each component contributes to generalization. These experiments reveal hierarchical dependencies between modules and demonstrate how multi-task learning prevents dataset-specific overfitting.

For computational efficiency, these experiments used a reduced training subset with evaluation on the corresponding validation set (PDBbind, BioLip, and ChEMBL targets – see Appendix D.3 for details.) As our goal was to assess the relative contribution and interdependency of the modules rather than to achieve maximum performance, this controlled environment is sufficient to reveal the core architectural dependencies, which we observed to emerge early in the training process.

Table 4 and Figure D2 show that MotifScreen's performance arises from synergy between modules rather than simple addition of components. The modules operate hierarchically: understanding physical geometry enables learning of abstract chemical patterns.

Table 4: AUROC performance ($\pm$ std) on validation set across ablation configurations at epoch 31. MotifScreen combines three modules: Affinity (Aff), Structure (Str), and Motif.

| Description | ChEMBL | PDBbind | BioLip | Params # |
|---|---|---|---|---|
| Aff + Str + Motif | **0.61 $\pm$ 0.14** | **0.63 $\pm$ 0.36** | **0.81 $\pm$ 0.31** | 2,472,702 |
| Aff + Motif | 0.54 $\pm$ 0.15 | 0.60 $\pm$ 0.35 | **0.81 $\pm$ 0.31** | 2,472,637 |
| Aff + Str | 0.59 $\pm$ 0.12 | 0.60 $\pm$ 0.37 | 0.79 $\pm$ 0.33 | 2,472,702 |
| Aff | 0.57 $\pm$ 0.18 | 0.55 $\pm$ 0.37 | 0.77 $\pm$ 0.34 | 2,472,637 |

**Impact of Structure Module.** The structure module learns physically plausible ligand geometry within receptor binding pockets. Removing this module (Aff + Motif configuration) reveals its importance for generalization. Performance remains stable on structure-rich PDBbind and BioLip datasets but drops significantly on ChEMBL ($0.62 \rightarrow 0.54$ AUROC).

This drop is significant because structure-based losses are computed only on PDBbind and BioLip, not ChEMBL. It demonstrates that geometric understanding acquired on structural targets transfers to ChEMBL targets lacking structural data, confirming the structure module's role as a generalizable feature extractor.

**Impact of Motif Module.** The motif module's contribution acts as a general-purpose enhancer, with its removal (Aff + Str configuration) causing consistent performance decrease across all datasets (Figure D2). This confirms that learning shared chemical motifs broadly enhances binder identification for both structural and non-structural datasets.

**Hierarchical Dependencies between Modules.** The motif module's effectiveness depends on the structure module. In the Aff + Motif configuration, motif learning helps when structure guidance is present but hurts ChEMBL performance when absent (drops below Aff baseline during training for ChEMBL targets, as shown in top right curve in Figure D2). This suggests that structure understanding enables meaningful motif learning while preventing spurious pattern recognition.

**Base Model Performance.** The base model (Aff only) provides a performance baseline, with consistently lower AUROC across all datasets and throughout training (Table 4, Figure D2, top left). This demonstrates that both structure module (physical validity and 3D geometry) and motif module (chemical interaction patterns) are essential, complementary components for robust virtual screening.

## 5 LIMITATIONS

While MotifScreen demonstrates promising results, several limitations should be acknowledged. First, our ablation experiments, though informative, were conducted on a reduced dataset subset with shorter training durations (31 epochs compared to full-scale training). This choice was made for computational tractability but inevitably limits the strength of conclusions regarding how different architectural components interact over longer training horizons. A more systematic, large-scale ablation study is therefore needed to disentangle the contributions of individual modules and auxiliary losses, and to assess whether similar synergies would emerge under full training conditions.

Second, a more granular analysis of the architecture could further optimize MotifScreen. Exploring architectural simplifications—such as removing or approximating more complex attention blocks—would help clarify which design choices are essential and whether comparable performance can be achieved with lower computational cost. In preliminary experiments, replacing certain components (e.g., using an EGNN instead of the SE(3) featurizer) reduced performance, which motivated us to retain the original design; nevertheless, alternative lightweight architectures may still offer promising directions for reducing cost in future work. In parallel, MotifScreen employs multiple objectives across three tasks, each with several associated losses. Systematically ablating these losses would reveal which are most critical for generalization and which are redundant.

Finally, because our model benefits from carefully curated and augmented data (e.g., activity filtering, cross-decoy construction), controlled data ablations will be important for formally decoupling dataset contributions from architectural ones. Data-centric approach is particularly crucial in biological do-

mains, where training data is often aggregated from multiple, multi-modal sources, each with inherent biases. Formally decoupling contributions of our data-centric philosophy from the architecture itself could reveal generalizable principles for combining complex data, with implications beyond virtual screening.

A limitation in our comparative study arises from the fundamental incompatibility between structure-based baselines and our training data. Methods like SurfDock require ground-truth 3D complex structures for all training samples, making it methodologically infeasible to retrain them on our large-scale ChEMBL split. This highlights a key advantage of MotifScreen: the ability to leverage massive non-structural bioactivity data, which existing structural baselines cannot utilize.

# 6    DISCUSSION

In this work, we addressed the critical challenge of over-optimistic performance in deep learning-based virtual screening, a problem rooted in the inherent biases and target leakage of widely used benchmarks. Our twofold contribution—the leakage-resistant ChEMBL-LR benchmark and the principle-guided MotifScreen model—provides a robust framework for developing and evaluating more generalizable virtual screening tools. Our experiments demonstrate that MotifScreen achieves more reliable and robust performance, particularly on ChEMBL-LR benchmark where existing VS models struggle.

We attribute this success to the synergy between our data curation strategy and architectural design. Multi-task learning may prevent overfitting by forcing the model to learn generalizable representations that satisfy multiple objectives simultaneously. Rather than optimizing solely for binding classification—which could lead to memorizing dataset-specific patterns—our framework requires the model to simultaneously understand physical geometry (structure module) and chemical interaction patterns (motif module). This architectural constraint encourages learning of fundamental protein-ligand interaction principles where physical validity guides the learning of chemical patterns. Our ablation studies support this hypothesis, revealing hierarchical dependencies where the structure module enables more effective motif learning.

The field of virtual screening has reached a critical juncture where the risk of inflated metrics on flawed benchmarks can stifle genuine progress. By demonstrating the significant performance drop of existing models on ChEMBL-LR and the resilience of MotifScreen, we not only validate our approach but also highlight the urgent need for a paradigm shift in how these models are trained and evaluated. We believe that the principles embodied in MotifScreen and the standards set by ChEMBL-LR offer a path toward more reliable and impactful *in silico* drug discovery.

## DATA AND MODEL AVAILABILITY (REPRODUCIBILITY STATEMENT)

To ensure full reproducibility, we have provided comprehensive details of our methodology: the model's architecture and training algorithm are detailed in Appendix C (Section C), all hyperparameters are listed in Appendix E (Section E.2), and the full dataset curation protocol is described in Appendix B (Section B.1). The computational setup and experimental details are provided in Appendix E (Section E). The final list of benchmark targets is available in Appendix F (Section F). The code for MotifScreen will be made publicly available on GitHub upon acceptance.

Furthermore, we will release the complete ChEMBL-LR benchmark, including the specific active and decoy assignments for each target, molecular structures, and evaluation splits. All baseline method implementations and evaluation protocols used in our comparisons will be documented and made available to facilitate fair future comparisons.

## ETHICS STATEMENT

We explicitly recognize the ethical importance of ensuring fair and unbiased evaluation in deep learning. As highlighted in our paper, the widespread use of biased benchmarks has led to inflated and misleading performance claims in virtual screening. To address this, we have curated a novel, leakage-proofed dataset with strict principles to eliminate data leakage and chemical bias. We

believe this contribution is essential for promoting research integrity and establishing a more rigorous foundation for future work in the field.

The authors have adhered to the ICLR Code of Ethics in all aspects of this research. A large language model was used to aid in refining the grammar of the manuscript; all scientific ideas, research questions, and experimental results are original to the authors.

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

# A ADDITIONAL RELATED WORKS

Here we provide a detailed overview of related works, with a specific focus on the conventional benchmarks and models used in virtual screening. The following sections aim to elucidate the characteristics and inherent weaknesses of these widely used datasets, providing context for the development of our ChEMBL-LR benchmark.

## A.1 CONVENTIONAL VIRTUAL SCREENING BENCHMARKS

Table A1 summarizes five commonly used benchmark datasets for virtual screening (VS).

- **MUV** (Rohrer & Baumann, 2009): The Maximum Unbiased Validation dataset was designed to evaluate ligand-based VS (LBVS) methods while minimizing analogue bias and artificial enrichment. Actives and decoys were drawn from PubChem BioAssay, filtered to remove assay artifacts, and curated into 18 targets (30 actives and 15,000 decoys each). Its splitting strategy focuses on separating actives in training and test, but does not account for all sources of ligand bias (i.e. just AA-AI in AVE bias).

- **DUD-E** (Mysinger et al., 2012): An enhanced version of the Directory of Useful Decoys. Actives are collected from ChEMBL (v09) and decoys from ZINC, chosen to be topologically dissimilar from actives while matched on simple physicochemical properties (e.g., molecular weight, logP, H-bond counts, net charge). Widely used for LBVS and SBVS evaluation, but highly susceptible to ligand memorization bias in the deep learning era.

- **DEKOIS 2.0** (Bauer et al., 2013): Designed to reduce bias for both LBVS and SBVS methods. Actives come from BindingDB, and decoys are chosen from ZINC using property-based matching scores (DOE, LADS). Provides 81 targets with 40 actives and 1,200 decoys each. Similar issues to DUD-E remain, as decoys are not context-dependent across receptors.

- **CASF-2016** (Su et al., 2018): A subset of PDBbind (v2016) designed for the Comparative Assessment of Scoring Functions (CASF). Contains 57 protein–ligand complexes and is used to benchmark docking, scoring, and ranking power. Unlike the datasets above, CASF focuses more on protein-ligand complex modeling rather than active/decoy discrimination. It uses PDBbind refined set as its only source without external database like ChEMBL or ZINC, so number of actives are limited per target.

- **LIT-PCBA** (Tran-Nguyen et al., 2020): The most recent among the five, released in DL era. Developed in direct response to biases in DUD-E and MUV. Actives and decoys are sourced from PubChem BioAssay, with decoys curated to minimize Asymmetric Validation Embedding (AVE) bias. Provides 15 protein targets with ∼500 actives and 27,000 decoys per target. Although effective for within-target splits, its limited number of targets and reliance on strict split assumptions restricts its use when external datasets are added to training.

**DUD-E** and **DEKOIS 2.0** decoys are selected to differ from actives primarily in global topology while matching basic physicochemical properties. However, they are assigned uniformly as non-binders across all receptors, regardless of biological context. In reality, a compound may act as an active ligand for one receptor while serving as a decoy for another. The lack of such context-dependent labeling makes the benchmark vulnerable to ligand memorization, since a model can exploit global chemical features without truly learning protein–ligand interactions.

## A.2 EXISTING MODELS: TRAINING AND TEST SETS

Here we rationalize our choice of baseline methods used for comparison.

**AutoDock Vina (ADV)** (Eberhardt et al., 2021) is a popular and accurate docking method that remain widely used. Many DL models for protein-ligand interaction prediction (both docking and screening) compare their performance against ADV.

**AK-Score2** (Hong et al., 2024) was selected as a representative scoring model. In their work, it showed better performance compared to other scoring functions such as RTMScore, while relying on docked protein-ligand structure as input. This reflects a common limitation of scoring models:

Table A1: Common benchmark datasets used for evaluation.

| Dataset | Targets | #Actives / #Decoys (per-target mean) | Sources |
|---|---|---|---|
| MUV (2012) | 18 | 30 / 15,000 | PubChem BioAssay |
| DUD-E (2012) | 102 | 654 / 13,926 | ChEMBL, ZINC |
| DEKOIS 2.0 (2013) | 81 | 40 / 1,200 | BindingDB, ZINC |
| CASF-2016 (2018) | 57 | 5 / 280 | PDBbind |
| LIT-PCBA (2020) | 15 | 523 / 27,092 | PubChem BioAssay |

their dependence on the docking tool used to generate complex structures, with ADV being the de facto choice. Following their paper, we generated protein-ligand complex structure using ADV for benchmarks.

**KarmaDock** (Zhang et al., 2023) represents a class of deep learning-based docking models designed also for virtual screening. Requires protein 3D structure with its binding pocket information and ligand smiles.

**SurfDock** (Cao et al., 2025) was chosen because it achieved a docking success rate of 78.4% in the benchmarking study of (Jiang et al., 2025), where it outperformed several other deep learning methods, including AlphaFold3 (Abramson et al., 2024) (60.3%). It also requires protein 3D structure as input.

Table A2 summarizes the training and evaluation datasets used by these methods. AK-Score2 attempted to remove DUD-E-like targets from their training set but used weak similarity thresholds: protein sequence identity >90% and ligand similarity >0.8, compared to our more stringent thresholds of 0.4 for both.

Table A2: Training and benchmark datasets used by existing methods and ours.

| Method | Training Set | Test Set | Redundancy |
|---|---|---|---|
| KarmaDock (Zhang et al., 2023) | PDBbind v2020 | DEKOIS 2.0 | None |
| SurfDock (Cao et al., 2025) | PDBbind v2020 | DEKOIS 2.0 | None |
| AK-Score2 (Hong et al., 2024) | PDBbind v2020 | CASF-2016, DUD-E | Weak removal[a] |
| **MotifScreen (Ours)** | PDBbind, BioLip, ChEMBL | ChEMBL-LR, DUD-E | Strict control[b] |

[a] Protein sequence identity >90%, ligand similarity >0.8
[b] Protein sequence identity >40%, ligand similarity >0.4, plus AVE bias control and target-level splitting

**Note on DUD-E benchmark.** Our comparative evaluation on DUD-E benchmark is deliberately stringent, as we avoid the target leakage that likely benefits published baseline results; this limits direct performance comparisons but provides a more honest measure of generalization.

# B ADDITIONAL METHODS: DATASET

## B.1 DATASET CONSTRUCTION

Our model's robust performance relies on a new training and benchmark dataset curated to overcome the biases prevalent in existing datasets. We constructed this dataset by combining publicly available protein-ligand structural and activity data from **PDBbind**, **BioLip**, and **ChEMBL**. Our curation process followed a series of strategies designed to ensure data quality and eliminate potential sources of leakage. This approach allows our model to learn generalizable principles of molecular interaction rather than memorizing biased features. As part of this work, we also propose a new benchmark dataset, **ChEMBL-LR**, as a rigorous benchmark for evaluating deep learning-based virtual screening models.

### B.1.1 DATA SOURCES AND PROCESSING

We utilized PDBbind, BioLip, and ChEMBL 34 to construct our training and validation sets, while the new benchmark set was derived exclusively from ChEMBL 34. Here, we detail the processing strategies for PDBbind and BioLip in detail.

**PDBbind** (Liu et al., 2017) and **BioLip** (Zhang et al., 2024): We extracted protein-ligand complex structures from these databases to train the structural components of our model. To maximize diversity, we included both the `refined` and `general` sets from PDBbind. The following processing strategies were applied:

- **Removal of Non-Drug-Like Molecules.** To ensure chemical relevance and data quality, we filtered the complexes to remove non-drug-like molecules. Ligands with more than 50 heavy atoms were excluded. Molecules that do not conform to drug-like characteristics, such as glycans and cofactors (e.g., nucleotides) along with other outliers identified by manual inspection, were removed. This step reduces chemical bias originating from non-pharmacological molecules in the source databases.

- **Sample Weighting via Clustering.** To mitigate redundancy from over-represented protein families, protein targets were clustered at a 50% sequence identity threshold. Each sample was then assigned a weight during training according to the formula $w = 2.0/\sqrt{n}$, where $n$ is the number of members in its respective cluster.

- **Cross-Docking Set Curation.** Building on the principle that homologous receptors often share active ligands due to comparable binding site chemistry, we designated ligands from highly similar proteins as putative actives for one another (see Section 3.3). This is based on the knowledge that homologous proteins commonly share binding site traits (Martin, 2010). A stringent sequence identity threshold of 95% was used to define receptor similarity, minimizing the risk of introducing false positive interactions that could be detrimental to model performance.

- **Decoy Selection.** Decoys were selected from molecules known to be active against other receptors (i.e., cross-decoys). To prevent the inclusion of false negatives, only molecules with Tanimoto similarity below 0.3 to the active were selected as decoys. Furthermore, to create a challenging discrimination task, decoys were matched to actives based on key physicochemical properties: molecular weight, LogP, topological polar surface area (TPSA), hydrogen bond donor/acceptor counts, and aromatic ring counts.

**ChEMBL 34.** (Gaulton et al., 2012)    We curated a large-scale activity dataset from ChEMBL to train our model's affinity prediction module. This dataset provided a vast and diverse set of compounds, including many without corresponding structural data, allowing our model to learn from a wider chemical space. To create our final benchmark, we partitioned the ChEMBL data. Targets with no homologous proteins in PDBbind or BioLip were allocated to the ChEMBL-LR benchmark set. The remaining ChEMBL targets were integrated into our main training set.

**Validation Set Splitting Strategy**    A robust validation set is crucial for monitoring training and preventing overfitting. To ensure strict separation between training and validation data, we performed two independent partitioning operations. First, all protein targets from PDBbind and BioLip were combined and split into their own training and validation sets using a 40% sequence identity threshold. Second, the same target-wise splitting procedure was separately applied to our ChEMBL training set. The final training set is composed of the training splits from both partitions, and the final validation set is composed of both validation splits. This strategy guarantees that no target in the validation set has a homologous protein within its corresponding training data pool. MMseqs2 (Kallenborn et al., 2024) was used to split targets.

### B.1.2 BENCHMARK CURATION STRATEGIES

To create a robust and unbiased benchmark for virtual screening, we partitioned the ChEMBL database into a dedicated training set and a novel test set, which we term ChEMBL-LR (Leakage-Resistant). The curation of the ChEMBL-LR benchmark was guided by the following strategies, designed to eliminate data leakage and bias and to ensure data quality:

- **Realistic Active Selection.** Active compounds were selected using an activity threshold that reflects a realistic virtual screening scenario. Specifically, we filtered for compounds with a $K_i$, $K_d$, $EC_{50}$ or $IC_{50}$ value below 10 µM. The resulting average affinity for our benchmark actives is 1.02 µM, contrasting with older benchmarks like DUD-E (0.035 µM average), which present an unrealistically simple task by using only highly potent binders (Tran-Nguyen et al., 2020).

- **Target Selection.** Also reflecting a realistic virtual screening scenario where binding site for target receptor is known (Zhou et al., 2024), we included only proteins witha single, well-defined pocket. To verify this, we searched the Protein Data Bank for all available ligand-bound crystal structures for each target (via its UniProt ID). Targets were retained only if all biologically relevant ligands (e.g. not a solvent) were observed to bind to the same pocket.

- **Prevention of Target-Based Leakage.** As previously described, targets with no homologous proteins in PDBbind or BioLip were allocated to the ChEMBL-LR benchmark set. The remaining ChEMBL targets were integrated into our main training set. This is an improvement upon datasets by (Tran-Nguyen et al., 2020), which minimized bias only through a within-target splitting procedure. Also becaues many DL methods use a subset of our training set, this dataset stands as a rigorous evaluation benchmark for previous methods as well.

- **Active Clustering to Reduce Scaffold Bias.** To mitigate sampling bias from over-represented chemical scaffolds in the ChEMBL database, we clustered the active compounds for each target. Using a Tanimoto distance threshold of 0.3, we selected only the compound with the highest affinity from each cluster as its representative.

- **Decoy Selection.** The decoy selection process followed the same property-matching protocol used for the PDBbind/BioLip sets. For the benchmark, decoys for a given target were drawn from the pool of curated actives for all other targets in ChEMBL. A total of 30 decoys were selected for each active.

- **Ligand-Based Bias Check.** We performed an Asymmetric Validation Embedding (AVE) bias analysis on the final training and test sets. This ensured that the chemical space of active and decoy molecules is homogeneously distributed, making it difficult for models to rely on simple ligand features.

### B.1.3 FINAL DATASET COMPOSITION

The final curated dataset consists of distinct sets for training, validation, and testing. PDBbind and BioLip structures were used exclusively for training and validation. From ChEMBL, we created a non-overlapping training set and our final test benchmark, **ChEMBL-LR**. For the final ChEMBL-LR benchmark, an additional manual inspection was performed to ensure structural quality, resulting in a selection of 60 high-quality targets. Final list of target UniProt IDs are at the end of this appendix (Appendix F).

The strict separation of training and testing data ensures that our benchmark is a rigorous and reliable measure of true model generalization. Table B1 provides a detailed summary of the number of active protein-ligand pairs in each of these sets.

Table B1: Protein-ligand (active) pairs in each set and their split.

| # P-L pairs | PDBbind | BioLip | ChEMBL |
|---|---|---|---|
| Training set | 14,988 | 33,405 | 40,918 |
| Validation set | 3,874 | 9,221 | 8,179 |
| Test set (ChEMBL-LR) | – | – | 6,382 |

Table B2: Training set used for benchmark on DUD-E and ChEMBL-LR

| Dataset | Kept | Excluded | |
|---|---|---|---|
| | | Similar Protein | Similar Protein-Ligand |
| PDBbind+BioLip (clustered) | 31,429 | 7,289 | 3,526 |

# C ADDITIONAL METHODS: MODEL DESIGN AND TRAINING

## C.1 OVERALL ARCHITECTURE

The end-to-end forward pass of MotifScreen, which is identical during training and inference, is detailed in Algorithm 1. The model takes protein receptor and ligand graphs as input and is trained to jointly predict binding motifs $\mathbf{M}$, ligand key atom coordinates $\mathbf{X}_{key}$, ligand key atom distance maps $\mathbf{D}_{key}$, normalized protein-ligand pair feature maps $\mathbf{Z}_{key}$ and binding scores $y$ along with score map $\mathbf{Aff}_{\text{contrast}}$.

---

**Algorithm 1** Forward pass of MotifScreen

1: **function** MOTIFSCREEN($G_{rec}, G_{lig}$)        ▷ Input: receptor and ligand graphs
2:   $\mathrm{h}_{grid}, \hat{\mathbf{M}} \leftarrow$ **GridFeaturizer**($G_{rec}$)
3:   $\mathrm{h}_{lig} \leftarrow$ **LigandFeaturizer**($G_{lig}$)
4:   $\mathrm{h}_{lig}^{global} \leftarrow$ **LigandEmbedding**($G_{lig}$)
5:   $\mathbf{D}_{grid}, \mathbf{D}_{lig} \leftarrow$ Encode pairwise distances
6:   $\mathbf{Z} \leftarrow$ **TrigonBlock**($\mathrm{h}_{grid}, \mathrm{h}_{lig}, \mathbf{D}_{grid}, \mathbf{D}_{lig}$)      ▷ initialize pairwise features
7:   $\mathrm{h}_{key} \leftarrow$ **Ligand-to-Key Mapping**($\mathrm{h}_{lig}$)
8:   $\mathrm{h}_{key} \leftarrow$ **Ligand-to-Key Attention**($\mathrm{h}_{key}, \mathrm{h}_{lig}$)
9:   $\mathbf{Z_{key}} \leftarrow$ **Key-Restricted Projection**($\mathbf{Z}$)
10:   **for** $\ell = 1 \ldots L$ **do**          ▷ Interaction (CrossTrigon) Module
11:    $\mathbf{D}_{key} \leftarrow$ **DistanceTransform**($\mathrm{h}_{key}$)
12:    $\mathrm{h}_{key} \leftarrow$ **CrossAttention**($\mathrm{h}_{key}, \mathrm{h}_{grid}, \mathbf{Z}_{key}$)
13:    $\mathrm{h}_{grid} \leftarrow$ **CrossAttention**($\mathrm{h}_{grid}, \mathrm{h}_{key}, \mathbf{Z}_{key}$)
14:    $\mathbf{Z}_{key} \leftarrow$ **TrigonBlock**($\mathrm{h}_{grid}, \mathrm{h}_{key}, \mathbf{D}_{grid}, \mathbf{D}_{key}$)
15:   **end for**
16:   $\hat{\mathbf{X}}_{key}, \hat{\mathbf{D}}_{key}, \hat{\mathbf{Z}}_{key} \leftarrow$ **StructureModule**($\mathbf{Z}_{key}, G_{rec}$)
17:   $\hat{y}, \hat{\mathbf{Aff}}_{\text{contrast}} \leftarrow$ **ScreeningModule**($\mathbf{Z}_{key}, \mathrm{h}_{grid}, \mathrm{h}_{key}, \mathrm{h}_{lig}^{global}$)
18:   **return** $\hat{\mathbf{M}}, \hat{\mathbf{X}}_{key}, \hat{\mathbf{D}}_{key}, \hat{\mathbf{Z}}_{key}, \hat{y}, \hat{\mathbf{Aff}}_{\text{contrast}}$
19: **end function**

---

## C.2 INPUT REPRESENTATION AND FEATURE ENGINEERING

To prepare the molecular data for our graph neural network, we convert both proteins and ligands into graph structures annotated with chemically relevant feature sets.

### C.2.1 GRAPH CONSTRUCTION

**Protein Graph.** A protein's binding pocket is represented by a graph whose nodes consist of the protein's heavy atoms and a set of virtual grid points. These grid points are intelligently placed to represent the interaction surface by first generating a dense grid around the binding site, then filtering it to remove clashing or distant points and retaining only the largest connected component. Edges in this graph are constructed based on spatial proximity, connecting each node to its k-nearest neighbors.

**Ligand Graph.** Similarly, the ligand graph's nodes consist of its atoms. To capture both covalent and non-covalent interactions, its edges are also constructed based on spatial proximity (k-nearest neighbors), not just the covalent bond topology.

**Ligand Key Atom Selection.** Key atoms are selected among the ligand heavy atoms prior to graph generation. Ligands are split into fragments following the rules for the breaking of retrosynthetically interesting chemical substructures (BRICS) Degen et al. (2008). BRICS fragments considers chemical environment of the cleavage bond and surrounding substructures Shao et al. (2024). A key atom is selected from each BRICS fragment to represent the substructure.

### C.2.2 NODE AND EDGE FEATURES

**Protein Node Features:** Each protein atom node is described by a vector containing a one-hot encoding of its amino acid type, a detailed atom type classification (e.g., distinguishing between different carbon hybridization states), Solvent-Accessible Surface Area (SASA), and partial charge.

**Grid Node Features.** In contrast to atoms with intrinsic properties, the virtual grid nodes act as spatial placeholders. They are initialized with only generic features, as it is the express purpose of the Motif Module (detailed in the following section) to predict and populate them with rich chemical information, effectively transforming the grid into an inferred map of the binding site's interaction potential.

**Ligand Node Features.** Each heavy atom in the ligand is represented by a feature vector that includes a one-hot encoding of its element type, normalized counts of neighboring atoms by bond type, SASA, an occlusion score, and its partial charge.

**Protein Edge Features.** Each edge in the protein graph (protein and grid node combined) is annotated with a vector describing the relationship between the two connected nodes, including a binary feature for covalent bonds, a binary feature for grid-grid connections, and a continuous feature derived from the Euclidean distance.

**Ligand Edge Features.** Each edge in the ligand's spatial graph is described by a feature vector encoding both covalent information (bond type, if the edge represents a bond) and topological distance (shortest path through the covalent bond graph).

### C.3 GENERATING PHYSICOCHEMICAL GUIDANCE FOR TRAINING

### C.3.1 FROM STRUCTURAL DATA: PDBBIND AND BIOLIP

**Motif Labels (M).** The ground truth for the Motif Module is derived from the physicochemical properties of the crystal ligand. We first identify the chemical nature of the ligand fragments (e.g., H-bond donor, acceptor, aromatic). These properties are then projected onto the nearby virtual grid points using a Gaussian decay function. This process creates a continuous field on the grid that represents the ideal chemical environment of the binding pocket, which the model is trained to predict.

**Structure Labels ($\mathbf{X}_{key}, \mathbf{D}_{key}$).** he ground truth for the Structure Module is taken directly from the crystal structure. The 3D coordinates of the ligand's pre-defined key atoms serve as the ground truth for the positioning task. The ground truth intra-ligand distance map is also calculated from these coordinates.

**Affinity Labels ($y$).** For the Affinity Module, known binding ligands (actives) are assigned a ground truth label of 1, while non-binding molecules (decoys) are assigned a label of 0.

### C.3.2 FROM ACTIVITY DATA: CHEMBL

For data from ChEMBL, we have protein structures and ligand SMILES, but no experimental complex structures. Therefore, ground truth is only available for the affinity task.

**Input Generation for ChEMBL.** Ligand 3D structures are generated from their SMILES strings using Open Babel. The corresponding protein structures are sourced from the PDB based on their UniProt ID. Structure with highest resolution is used.

**Affinity Labels ($y$).** The ground truth is determined by the reported bioactivity. Compounds meeting the activity threshold are labeled as actives (1), and all others (decoys) are labeled as inactives (0).

**Guidance for ChEMBL Training.**     Crucially, for ChEMBL data, there is no ground truth available for the Motif and Structure modules. During training, the loss functions for these two modules are simply not computed for samples originating from ChEMBL. This allows the model to learn from large-scale activity data to improve its affinity prediction, while still learning the principles of geometry and chemical motifs from the high-quality structural data.

## C.4    MODEL COMPONENTS

### C.4.1    SE(3)-EQUIVARIANT FEATURIZERS (GRIDFEATURIZER, LIGANDFEATURIZER)

The initial processing of protein and ligand graphs is handled by SE(3)-equivariant featurizers, which generate rotationally and translationally equivariant representations—a crucial inductive bias for molecular data.

**GridFeaturizer.**     The protein binding pocket, represented by both atomic and virtual grid nodes, is processed by an SE(3)-Transformer. This featurizer leverages higher-order spherical harmonics to capture complex geometric relationships, which our ablations (replacing SE(3)-Transformer to simpler EGNNs) suggest is critical for learning meaningful motifs. The output is a scalar feature vector for each node, $h_{rec}$, and a set of motif predictions $\hat{M}$ for the grid points.

**LigandFeaturizer.**     The ligand graph is processed by a Graph Attention Network (GAT), which updates each atom's features based on its neighbors. This produces a scalar feature vector, $h_{lig}$.

### C.4.2    INTERACTION MODULE

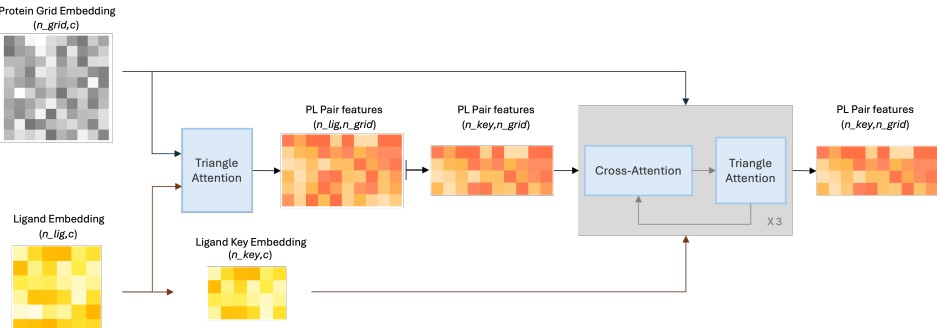

Figure C1: **Detailed Architecture of the Interaction Module.** This figure illustrates the iterative refinement process within the Interaction Module. It transforms initial protein grid and ligand key atom embeddings into a rich, geometrically-informed protein-ligand pair feature map. The module consists of alternating Cross-Attention (for inter-molecular information exchange) and Triangle Attention blocks (for intra- and inter-molecular geometric reasoning), leveraging distance maps (not depicted) to enhance feature propagation.

The core of MotifScreen is an Interaction Module (Figure C1) that iteratively refines the protein and ligand representations over L layers. The information flow is designed as a sequential, iterative process where node and pair features are updated in tandem. For any given layer l in the stack, the operation proceeds as follows, as detailed in Algorithm 2:

1. **DistanceTransform.** Projects ligand key atom node embeddings, $h_{key}^{(l-1)}$, into a latent space and computes a normalized outer-product. This produces a learned pairwise distance $\mathbf{D}_{key}$ which allows the model to represent flexible intra-ligand relationships beyond static Euclidean distances.

2. **CrossAttention.** Updates the node embeddings. As detailed in Algorithm 3, this step uses the pairwise feature map from the previous layer, $\mathbf{Z}_{key}^{(l-1)}$, to weight and guide the information exchange between protein grid points and ligand key atoms. This produces updated node embeddings for the current layer, $h_{grid}^{(l)}$ and $h_{key}^{(l)}$. CrossAttention allows the protein representation to be updated based on the ligand's features, and vice-versa.

3. **TrigonBlock.** Refines the pair feature map. As shown in Algorithm 4, this block takes the newly updated node embeddings, $h_{grid}^{(l)}$ and $h_{key}^{(l)}$, and their respective distance maps ($\mathbf{D}_{grid}$ and the learned $\mathbf{D}_{key}$) to compute a new, more refined pairwise interaction tensor, $\mathbf{Z}_{key}^{(l)}$. This is achieved through a sequence of Triangle Multiplicative Updates and Triangle Attention layers following (Jumper et al., 2021), allowing the model to reason about higher-order geometric relationships.

The outputs of this layer ($h^{(l)}$ and $\mathbf{Z}_{key}^{(l)}$) are passed as inputs to the next layer, $l + 1$, enabling a deep, iterative refinement of the protein-ligand complex representation.

---

**Algorithm 2** Interaction module (CrossTrigon)

---

1: **function** CROSSTRIGON($h_{grid}, h_{key}, \mathbf{Z}_{key}, \mathbf{D}_{grid}, L$)
2:     **for** $\ell = 1 \dots L$ **do**
3:         $\mathbf{D}_{key} \leftarrow$ **DistanceTransform**($h_{key}$)
4:         $h_{key} \leftarrow$ **CrossAttention**($h_{key}, h_{grid}, \mathbf{Z}_{key}$)
5:         $h_{grid} \leftarrow$ **CrossAttention**($h_{grid}, h_{key}, \mathbf{Z}_{key}$)
6:         $\mathbf{Z}_{key} \leftarrow$ **TrigonBlock**($h_{grid}, h_{key}, \mathbf{D}_{grid}, \mathbf{D}_{key}$)
7:     **end for**
8:     **return** $h_{grid}, h_{key}, \mathbf{Z}_{key}$
9: **end function**

---

**Algorithm 3** CrossAttention

---

1: **function** CROSSATTENTION($V, Q, \mathbf{Z}, \mathbf{Z}_{mask}$)
2:     $\tilde{\mathbf{Z}} \leftarrow \exp(\mathbf{Z}) \odot \mathbf{Z}_{mask}$                                        $\triangleright$ apply mask
3:     $\tilde{\mathbf{Z}} \leftarrow \tilde{\mathbf{Z}}/\sum \tilde{\mathbf{Z}}$               $\triangleright$ normalize across attended dim
4:     $Q_a \leftarrow \text{Linear}_1(Q)$
5:     $V_a \leftarrow \tilde{\mathbf{Z}} \cdot Q_a$                $\triangleright$ aggregate queries into values
6:     $V \leftarrow V + \text{Linear}_2(V_a)$               $\triangleright$ residual update
7:     **return** $V$
8: **end function**

---

**Algorithm 4** Trigon block

---

1: **function** TRIGONBLOCK($h_{grid}, h_{key}, \mathbf{D}_{grid}, \mathbf{D}_{key}$)
2:     $h'_{grid} \leftarrow \text{Linear}(h_{grid}), \quad h'_{key} \leftarrow \text{Linear}(h_{key})$
3:     $\mathbf{Z} \leftarrow h'_{grid} \otimes h'_{key}$              $\triangleright$ initialize pair features
4:     **for** $t = 1 \dots n$ **do**
5:         $\mathbf{Z} \leftarrow \mathbf{Z} + $ **TriangleMultiplicativeUpdate**($\mathbf{Z}, \mathbf{D}_{grid}, \mathbf{D}_{key}$)
6:         $\mathbf{Z} \leftarrow \mathbf{Z} + $ **TriangleAttentionRowWise**($\mathbf{Z}$)
7:         $\mathbf{Z} \leftarrow \mathbf{Z} + $ **FeedForward**($\mathbf{Z}$)
8:     **end for**
9:     **return** $\mathbf{Z}$
10: **end function**

---

### C.4.3 PREDICTION HEADS

The final, refined embeddings are passed to separate modules for the multi-task predictions.

- **Structure Module.** This module takes the final pairwise features $\mathbf{Z}_{key}$ and predicts the 3D coordinates of the ligand key atoms ($\mathbf{X}_{key}$) and a predicted intra-ligand distance map ($\mathbf{D}_{key}$). It also returns the attention map ($\mathbf{A}$).
- **Screening Module.** The final screening module (Algorithm 6) processes the refined pairwise and node embeddings to produce two outputs: a scalar binding score $\hat{y}$ for binder classification and a per-key-atom affinity map $\mathbf{Aff}_{contrast}$ used for contrastive learning. The final score is a weighted average of a term derived from the protein-ligand interaction features and a term from the global ligand features.

---

**Algorithm 5** Structure Module for Key Atom Positioning

---

1: **function** STRUCTUREMODULE($\mathbf{Z}_{key}, G_{rec}$)
2:     $\mathbf{X}_{grid} \leftarrow G_{rec}$                          ▷ Get grid coordinates from receptor graph input.
3:     $\mathbf{A}_{logits} \leftarrow \text{Linear}(\mathbf{Z}_{key})$
4:     $\mathbf{A} \leftarrow \text{masked\_softmax}(\mathbf{A}_{logits}, \text{mask} = \mathbf{M}_z, \dim = 1)$
5:                                    ▷ Compute attention weights over grid points for each key atom.
6:     $\hat{\mathbf{X}}_{key,j} \leftarrow \sum_{i=1}^{N_{grid}} \mathbf{A}_{i,j} \cdot \mathbf{X}_{grid,i}$    for each key atom $j$
7:     **return** $\hat{\mathbf{X}}_{key}, \mathbf{A}$
8: **end function**

---

**Algorithm 6** Screening module

---

1: **function** SCREENINGMODULE($\mathbf{Z}_{key}, \text{h}_{grid}, \text{h}_{key}, \text{h}_{lig}^{global}, w_{mask}$)
2:     $\text{h}_{grid} \leftarrow \text{Softmax}(\text{h}_{grid}, \text{channel})$
3:     $\text{h}_{key} \leftarrow \text{Softmax}(\text{h}_{key}, \text{channel})$
4:     $\mathbf{a} \leftarrow \text{normalize}(\text{Affmap})$
5:     $\hat{\mathbf{Aff}}_{contrast} \leftarrow \text{h}_{key} \cdot \mathbf{a}$                            ▷ per-key contrastive scores
6:     $\text{key\_P} \leftarrow \sum_d Z_{:,:,k,d}\, h_{:,k,d}^{key}$
7:     $\text{aff}_{key} \leftarrow \max_n \text{key\_P}_{n,k}$                         ▷ max over grid
8:     $\text{aff}_{key} \leftarrow \text{Linear}(\text{aff}_{key})$                        ▷ scale + offset
9:     $\text{aff}_{key} \leftarrow \text{MaskedAverage}(\text{aff}_{key}, w_{mask})$
10:    $\text{aff}_{lig} \leftarrow \text{Linear}(\text{h}_{lig}^{global})$
11:    $\hat{y} \leftarrow \dfrac{\text{aff}_{key} + \Gamma \cdot \text{aff}_{lig}}{1 + \Gamma}$
12:    **return** $\hat{y}, \hat{\mathbf{Aff}}_{contrast}$
13: **end function**

---

## C.5 Multi-task Loss functions

MotifScreen is trained end-to-end with a composite objective function, which is a weighted sum of losses from our three primary tasks: 1) Motif classification losses, 2) Structural losses, and 3) Screening Losses. Each component in defined below. The full training loop, including loss calculation, is shown in Algorithm 7.

**Motif classification losses.**

- **Masked BCE:** positive and negative binary cross-entropy terms applied to motif predictions restricted by valid masks.
- **Contrastive attention loss:** auxiliary penalty encouraging attention overlap with correct motif positions.

**Structural losses.**

- **Structure loss:** regression on key atom coordinates using MSE or Huber loss.
- **Pair distance loss:** hybrid cross-entropy + Huber objective matching predicted vs. true pairwise distances.
- **Spread loss:** attention regularization between predicted grid-key interaction map $\hat{\mathbf{Z}}_{key}$ and ground-truth ligand coordinates. It consists of two terms: (a) a positive alignment term encouraging overlap with Gaussian kernels at ligand positions ($\mathcal{L}_{\text{spread-pos}}$), and (b) a deviation penalty discouraging attention far from ligand positions ($\mathcal{L}_{\text{spread-neg}}$). The final contribution is $\mathcal{L}_{\text{str-spread}} = w_{\text{spread}}(\mathcal{L}_{\text{spread-pos}} + 0.2 \cdot \mathcal{L}_{\text{spread-neg}})$.

**Screening losses.**

- **Screening loss:** binary cross-entropy with logits on the scalar binding prediction $\hat{y}$.
- **Ranking loss:** KL divergence between predicted and label affinity distributions.
- **Screening contrast loss:** $L_2$ penalty aligning the per-key affinity map $\mathbf{Aff}_{contrast}$ with global binding labels. Enforces that the per-key affinity map for actives approaches all-ones, while for decoys it approaches all-zeros.

**Final objective.** The weighted objective is

$$\mathcal{L} = w_{\text{motif}}\big(\mathcal{L}_{\text{motif-pos}} + \mathcal{L}_{\text{motif-neg}} + \mathcal{L}_{\text{motif-contrast}} + w_{\text{motif-penalty}} \cdot \mathcal{L}_{\text{motif-penalty}}\big)$$
$$+ w_{\text{str}}\big(\mathcal{L}_{\text{str-dist}} + \mathcal{L}_{\text{str-pair}} + \mathcal{L}_{\text{str-attmap}}\big)$$
$$+ w_{\text{screen}} \cdot \mathcal{L}_{\text{screen}} + w_{\text{screen-rank}} \cdot \mathcal{L}_{\text{screen-rank}} + w_{\text{screen-contrast}} \cdot \mathcal{L}_{\text{screen-contrast}} \tag{1}$$
$$+ w_{\text{penalty}} \cdot \mathcal{L}_2. \tag{2}$$

Parameter regularization and quadratic penalty on motif attention magnitude to prevent collapse is not shown in the equation but is a part of training loss. All weights $w$ are hyperparameters set in the configuration (Table E2), and a per-target multiplier is applied to balance data imbalance across targets.

## C.6 TRAINING OF MOTIFSCREEN

Training was done using the dataset in Appendix B.1.1, Table B1. During training, each batch includes single target with 1 active and 5 decoys (decoys are randomly sampled among pre-selected decoy list). Model parameters for main benchmark was selected using metrics calculated on validation set. Detailed computational set-up is at Appendix E.

Algorithm 7 details the training loop executed in each epoch.

---

**Algorithm 7** Training loop for one epoch

---

1: **for** each batch $(G_{rec}, G_{lig}, \text{labels})$ **do**
2:     $(\hat{\mathbf{X}}_{key}, \hat{\mathbf{D}}_{key}, \hat{\mathbf{Z}}_{key}, \hat{\mathbf{M}}, \hat{y}, \hat{\mathbf{Aff}}_{contrast}) \leftarrow \textbf{MotifScreen}(G_{rec}, G_{lig})$
3:     Initialize all loss terms to zero
                                                                    ▷ — Motif classification losses —
4:     **if** motif labels available **then**
5:         $\mathcal{L}_{\text{motif-pos}}, \mathcal{L}_{\text{motif-neg}} \leftarrow \textbf{MaskedBCE}(\hat{\mathbf{M}}, \mathbf{M})$
6:         $\mathcal{L}_{\text{motif-contrast}} \leftarrow \textbf{ContrastLoss}(\hat{\mathbf{M}})$
7:         $\mathcal{L}_{\text{motif-penalty}} \leftarrow \text{ReLU}\left(\|\hat{\mathbf{M}}\|^2 - 25.0\right)$
8:     **end if**
                                                                            ▷ — Structural losses —
9:     **if** structure labels available **then**
10:        $\mathcal{L}_{\text{str-dist}}, \text{MAE} \leftarrow \textbf{StructureLoss}(\hat{\mathbf{X}}_{key}, \mathbf{X}_{key})$
11:        $\mathcal{L}_{\text{str-pair}} \leftarrow \textbf{PairDistanceLoss}(\hat{\mathbf{D}}_{key}, \mathbf{X}_{key})$
12:        $\mathcal{L}_{\text{str-spread}} \leftarrow \textbf{SpreadLoss}(\mathbf{X}_{key}, \hat{\mathbf{Z}}_{key})$
13:    **end if**
                                                                            ▷ — Screening losses —
14:    $\mathcal{L}_{\text{screen}} \leftarrow \textbf{BCEWithLogits}(\hat{y}, y)$
15:    $\mathcal{L}_{\text{screen-rank}} \leftarrow \textbf{RankingLoss}(\sigma(\hat{y}), y)$
16:    $\mathcal{L}_{\text{screen-contrast}} \leftarrow \textbf{ScreeningContrastLoss}(\hat{\mathbf{Aff}}_{contrast}, y)$
                                                                                ▷ — Regularization —
17:    $\mathcal{L}_2 \leftarrow \sum_{\theta \in \text{params}} \|\theta\|_2$
                                                                            ▷ — Final objective —
18:    $\mathcal{L} \leftarrow w_{\text{motif}} \cdot (\mathcal{L}_{\text{motif-pos}} + \mathcal{L}_{\text{motif-neg}} + \mathcal{L}_{\text{motif-contrast}} + w_{\text{motif-penalty}} \cdot \mathcal{L}_{\text{motif-penalty}})$
19:        $+ w_{\text{str}} \cdot (\mathcal{L}_{\text{str-dist}} + \mathcal{L}_{\text{str-pair}} + \mathcal{L}_{\text{str-spread}})$
20:        $+ w_{\text{screen}} \cdot \mathcal{L}_{\text{screen}} + w_{\text{screen-rank}} \cdot \mathcal{L}_{\text{screen-rank}} + w_{\text{screen-contrast}} \cdot \mathcal{L}_{\text{screen-contrast}}$
21:        $+ w_{\text{penalty}} \cdot \mathcal{L}_2$
22:    Backpropagate and update parameters with optimizer
23: **end for**

---

## C.7 KEY NOTATION

All mathematical symbols used in this work, as well as those appearing in algorithms and functions, are listed in Table C1.

Table C1: Notation used in the model and training algorithms.

| Symbol | Description |
|---|---|
| **Input Graphs and Embeddings** | |
| $G_{rec}, G_{lig}$ | Receptor and ligand input graphs |
| $\mathrm{h}_{rec}, \mathrm{h}_{grid}$ | Receptor atom and grid point node embeddings |
| $\mathrm{h}_{lig}, \mathrm{h}_{key}$ | Ligand and key atom node embeddings |
| $\mathrm{h}_{lig}^{global}$ | Global ligand embedding capturing holistic properties |
| $\mathbf{Z}, \mathbf{Z}_{key}$ | Pairwise interaction features between protein-ligand nodes |
| $\mathbf{D}_{grid}, \mathbf{D}_{key}$ | Pairwise distance map between grid points and key atoms |
| **Model Predictions** | |
| $\hat{\mathbf{M}}$ | Predicted motif probabilities on grid points |
| $\hat{\mathbf{X}}_{key}$ | Predicted 3D coordinates of ligand key atoms |
| $\hat{\mathbf{D}}_{key}$ | Predicted pairwise distance map between key atoms |
| $\hat{\mathbf{Z}}_{key}$ | Final normalized pairwise feature map |
| $\hat{y}$ | Final scalar binding prediction (screening score) |
| $\hat{\mathbf{Aff}}_{contrast}$ | Per-key-atom affinity map for contrastive loss |
| **Ground Truth and Hyperparameters** | |
| $\mathbf{X}_{key}$ | Ground truth 3D coordinates of ligand key atoms |
| $\mathbf{M}$ | Ground truth motif label of grid points |
| $y$ | Binary binding label (1: active, 0: decoy) |
| $L$ | Number of layers in the Interaction Module |
| $w.$ | Loss weights set in the configuration |

# D FURTHER MODEL ANALYSIS

## D.1 DETAILED MOTIFSCREEN PERFORMANCE

This section provides a detailed statistical breakdown of MotifScreen's performance on the two primary benchmarks evaluated: our proposed ChEMBL-LR and the external DUD-E benchmark.

Table D1 summarizes the performance on ChEMBL-LR. Beyond the mean AUROC and EF1%, it includes other metrics like BEDROC and an analysis of the predicted score distributions for active and decoy compounds. MotifScreen achieves a statistically significant separation between active and decoy scores ($p < 0.05$) for 72% of the targets on this benchmark.

Figure D1 further illustrates MotifScreen's robustness on ChEMBL-LR. The threshold curves show that our model sustains higher performance across a greater number of targets compared to baselines, a nuance not fully captured by mean values alone.

Table D2 provides the corresponding performance summary on the DUD-E benchmark.

Table D1: MotifScreen Performance Summary on the ChEMBL-LR Benchmark

| Metric | Mean $\pm$ Std | Median | Min | Max |
|--------|---------------|--------|-----|-----|
| AUROC | $0.680 \pm 0.165$ | 0.663 | 0.180 | 0.961 |
| EF1% | $4.16 \pm 5.65$ | 1.82 | 0.00 | 23.08 |
| BEDROC | $0.146 \pm 0.205$ | 0.086 | -0.077 | 0.811 |

Targets with significantly higher active vs. decoy scores ($p < 0.05$): **72%**

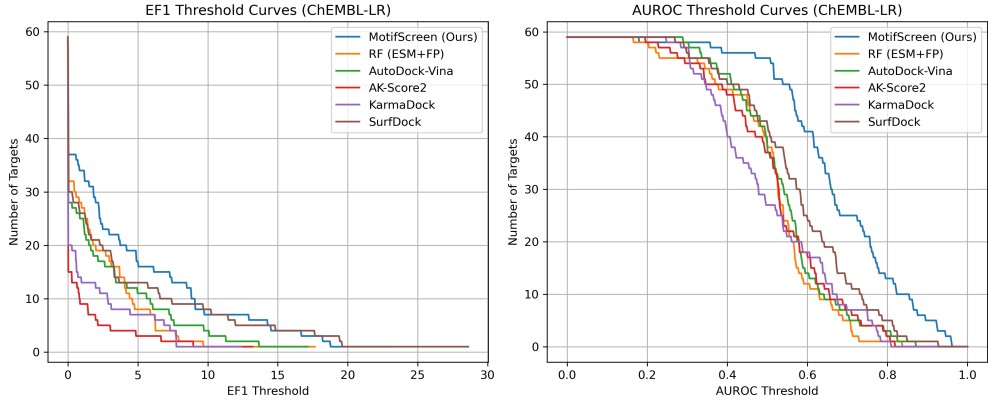

Figure D1: **Comparative EF1% and AUROC threshold curves on ChEMBL-LR.** Threshold analysis of screening performance on ChEMBL-LR. Left: number of targets with EF1% above a given threshold. Right: number of targets with AUROC above a given threshold. MotifScreen sustains higher performance across more targets, reflecting its robust performance on ChEMBL-LR.

Table D2: MotifScreen Performance Summary on the DUD-E Benchmark

| Metric | Mean $\pm$ Std | Median | Min | Max |
|--------|---------------|--------|-----|-----|
| AUROC | $0.753 \pm 0.137$ | 0.771 | 0.365 | 0.989 |
| EF1% | $5.94 \pm 5.47$ | 4.34 | 0.00 | 26.98 |
| BEDROC | $0.344 \pm 0.261$ | 0.329 | -0.121 | 0.920 |

Targets with significantly higher active vs. decoy scores ($p < 0.05$): **90%**

## D.2 Detailed Comparison with Baseline Methods

To provide full context for the results presented in the main text, this section includes detailed statistical tests and a comprehensive cross-benchmark comparison.

Tables D3 and D4 report pairwise statistical significance tests (Wilcoxon signed-rank) comparing MotifScreen against all baseline methods on the ChEMBL-LR benchmark. For AUROC, Motif-Screen is significantly better than all baselines (BH-corrected $p < 10^{-4}$). For EF1%, MotifScreen significantly outperforms AutoDock-Vina, KarmaDock, AK-Score2, and RF ($p < 0.05$). While the improvement over SurfDock is may not seem statistically significant just from this value, we have shown in Section 4.2.1 and Figure D1, that MotifScreen sustains better performance across more targets.

Finally, Table D5 presents the absolute AUROC and EF1% scores for all methods across the ChEMBL-LR, DUD-E, and DEKOIS 2.0 benchmarks. This table contains the full data used to calculate the performance drop ($\Delta$) values reported in the main body of the paper.

Table D3: Pairwise statistical significance tests (AUROC, Wilcoxon signed-rank) on ChEMBL-LR benchmark.

| Comparison | N pairs | Mean Diff (MS–X) | p-value | BH-adj p |
|---|---|---|---|---|
| MS vs AutoDock-Vina | 60 | +0.139 | $8.4 \times 10^{-9}$ | $7.6 \times 10^{-8***}$ |
| MS vs AK-Score2 | 60 | +0.152 | $1.5 \times 10^{-6}$ | $2.7 \times 10^{-6***}$ |
| MS vs KarmaDock | 60 | +0.166 | $8.9 \times 10^{-14}$ | $1.6 \times 10^{-12***}$ |
| MS vs SurfDock | 60 | +0.103 | $5.3 \times 10^{-5}$ | $7.9 \times 10^{-5***}$ |
| MS vs RF (ESM+FP) | 60 | +0.162 | $1.6 \times 10^{-14}$ | $3.1 \times 10^{-13***}$ |

[*] MS denotes MotifScreen.
[*] $p < 0.05$, [**] $p < 0.01$, [***] $p < 0.001$ (BH corrected).

Table D4: Pairwise statistical significance tests (EF1%, Wilcoxon signed-rank) on ChEMBL-LR benchmark.

| Comparison | N pairs | Mean Diff (MS–X) | p-value | BH-adj p |
|---|---|---|---|---|
| MS vs AutoDock-Vina | 60 | +1.97 | 0.0250 | $0.0301^{*}$ |
| MS vs AK-Score2 | 60 | +3.36 | $4.1 \times 10^{-5}$ | $2.5 \times 10^{-4***}$ |
| MS vs KarmaDock | 60 | +2.84 | $4.6 \times 10^{-4}$ | $0.00139^{**}$ |
| MS vs SurfDock | 60 | +0.72 | 0.161 | 0.161 |
| MS vs RF (ESM+FP) | 60 | +2.04 | 0.0162 | $0.0243^{*}$ |

[*] MS denotes MotifScreen.
[*] $p < 0.05$, [**] $p < 0.01$, [***] $p < 0.001$ (BH corrected).

## D.3 Ablation Study Details

### D.3.1 Dataset and Training

**Dataset.** To ensure computational tractability, all ablation studies were conducted on a reduced subset of our main training data. This subset was curated from the original sources as follows, reducing the total number of protein-active pairs from 89,331 to 58,773 for training and from 21,274 to 15,457 for validation.

- **PDBbind & BioLip:** All unique targets from these sources in our full training set were included.
- **ChEMBL:** A representative subset of 233 targets was selected by: 1) clustering all ChEMBL training targets at a 40% sequence identity threshold; 2) filtering out clusters corresponding to targets with fewer than 30 active compounds; and 3) selecting one representative target from each remaining cluster using a fixed random seed for reproducibility.

Table D5: AUROC and EF1% across ChEMBL-LR, DUD-E, and DEKOIS 2.0.

| Model | ChEMBL-LR | | DUD-E | | DEKOIS 2.0[c] | |
|---|---|---|---|---|---|---|
| | AUROC | EF1% | AUROC | EF1% | AUROC | EF1% |
| AutoDock-Vina (2021)[a] | $0.541 \pm 0.125$ | $2.189 \pm 3.439$ | 0.72 | 9.70 | 0.633 | 4.513 |
| AK-Score2 (2024)[b] | $0.527 \pm 0.135$ | $0.803 \pm 2.308$ | - | 14.6 | - | - |
| KarmaDock (2023) | $0.512 \pm 0.124$ | $1.317 \pm 2.715$ | 0.754 | 15.873 | 0.782 | 15.83 |
| SurfDock (2025) | $0.576 \pm 0.151$ | $3.443 \pm 6.076$ | - | - | 0.758 | 18.17 |
| RF (ESM+FP) | $0.518 \pm 0.131$ | $2.09 \pm 3.20$ | 0.691 | 14.21 | - | - |
| **MotifScreen (Ours)** | $\mathbf{0.680 \pm 0.165}$ | $\mathbf{4.16 \pm 5.65}$ | 0.753 | 5.94 | 0.63 | 3.18 |

[a] Reported in (Eberhardt et al., 2021).
[b] AK-Score2-DockC variant, results from (Hong et al., 2024).
[c] DEKOIS 2.0 results adopted from (Cao et al., 2025).
   Note: -" indicates the dataset was not evaluated by the original authors.

All ablation models were trained for 31 epochs, whereas the final model was trained for 120 epochs. As mentioned in Experiments (Section 4.3), we observed the core architectural dependencies to emerge early in the training process. Except for the reduced training duration, the experimental setup (hyperparameters, loss functions, and batching strategy) was identical to the protocol used for the full MotifScreen model. All hyperparameters, including learning rate, optimizer settings, and batch size, were held constant across all four experiments to ensure a fair comparison.

### D.3.2 MODEL CONFIGURATIONS

We evaluated four distinct model configurations to systematically assess the contribution of different components:

1. **Full Model:** The complete MotifScreen architecture with all modules and loss functions.

2. **No Structure Module:** The structure module and its associated losses were removed.

3. **No Motif Module:** The motif module and its associated losses removed.

4. **Base Model:** Both the structure and motif modules (and their associated losses) removed, leaving only the core affinity prediction components.

Evaluation was performed on the validation set corresponding to our reduced training data (see Section B.1 for splitting details). This approach isolates the effects of the ablated components under identical, controlled conditions. These results are intended to demonstrate the relative contribution of each module, not to serve as a direct performance comparison with the fully-trained model on the ChEMBL-LR benchmark.

### D.3.3 ADDITIONAL FIGURES

This section includes additional AUROC curves showing results for ablation studies (Figure D2)

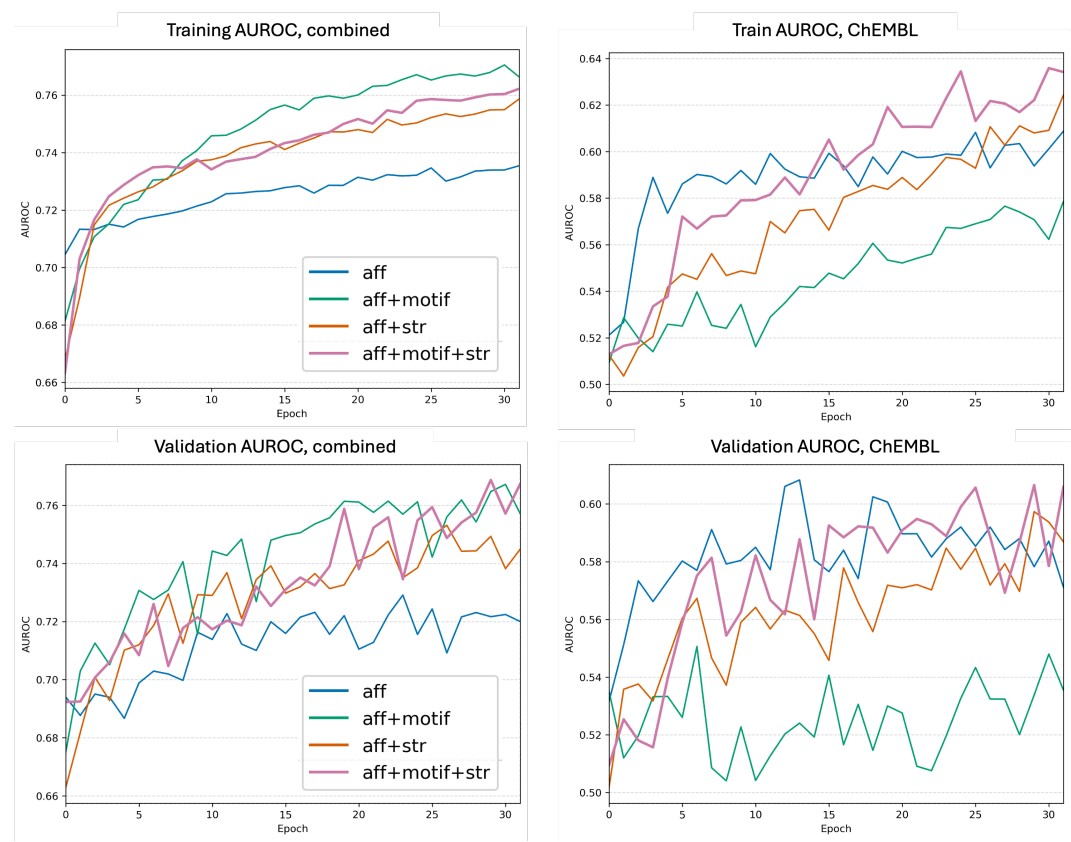

Figure D2: **Ablation study performance analysis.** Showing AUROC progression over epochs for training and validation sets. The complete MotifScreen (Aff+Str+Motif) achieves best performance.

# E  EXPERIMENTAL DETAILS

## E.1  COMPUTATIONAL SETUP

The MotifScreen model was implemented in PyTorch and used the Deep Graph Library (DGL) for graph operations. Training was conducted on a server with four NVIDIA A6000 GPUs using the Distributed Data Parallel (DDP) backend.

Inference is highly efficient. When running on a four NVIDIA A4000 GPUs with a batch size of 5, MotifScreen takes approximately 0.03 seconds per compound. SurfDock (Cao et al., 2025), which showed second to best performance on ChEMBL-LR, takes 10 to 11 seconds per compound. On a single CPU core, it takes approximately 1 second per compound.

## E.2  HYPERPARAMETERS

TableE1 and E2 list the key hyperparameters used for the MotifScreen model architecture and training process.

Table E1: Model Architecture Hyperparameters.

| **General** | |
| --- | --- |
| Dropout Rate | 0.2 |
| **Featurizers** | |
| GridFeaturizer Layers | 5 |
| LigandFeaturizer Layers | 4 |
| Input Scalar Features (Grid/Ligand) | 102 / 18 |
| Output Features (Grid/Ligand) | 64 / 64 |
| Attention Heads | 4 |
| Hidden Channels | 32 |
| Global Embedding Dim (Input/Output) | 19 / 4 |
| **Trigon / CrossTrigon Modules** | |
| Embedding Channels ($c$) | 64 |
| Distance Channels ($d$) | 64 |
| Ligand–Grid Trigon Layers | 2 |
| Key-level Trigon Layers | 3 |

Table E2: Training and Data Hyperparameters.

| Loss Weights | |
| --- | --- |
| $w_{\text{motif}}$ | 0.05 |
| $w_{\text{motif-penalty}}$ | $1 \times 10^{-10}$ |
| $w_{\text{str}}$ | 0.2 |
| $w_{\text{screen}}$ | 0.5 |
| $w_{\text{screen-contrast}}$ | 0.5 |
| $w_{\text{screen-ranking}}$ | 5.0 |
| $w_{\text{penalty}}$ | $1 \times 10^{-5}$ |
| Structure Loss Type | MSE |
| Screening Loss Type | BCE |
| **Optimization** | |
| Learning Rate | $1 \times 10^{-4}$ |
| Optimizer Weight Decay | $1 \times 10^{-4}$ |
| Max Epochs | 120 |
| Batch Size | 1 |
| Gradient Accumulation Steps | 1 |
| **Graph Preprocessing** | |
| Edge Mode | top-$k$ with $k$=8 |
| Max Edges / Nodes | 35,000 / 3,000 |
| Randomization (Grid / Data) | 0.5 / 0.2 |

# F ChEMBL-LR Target List

The 60 targets comprising the ChEMBL-LR benchmark are shown in Table E3 by their UniProt IDs with number of active compounds and assigned decoy compounds.

Table E3: Number of actives and decoys for the 60 ChEMBL-LR targets

| Target (UniProt ID) | Actives | Decoys | Target (UniProt ID) | Actives | Decoys |
|---|---|---|---|---|---|
| B2RXC2 | 17 | 510 | Q07794 | 56 | 1680 |
| O00429 | 16 | 480 | Q10469 | 61 | 1830 |
| O14920 | 369 | 11070 | Q13418 | 84 | 2520 |
| O15111 | 178 | 5340 | Q14524 | 264 | 7920 |
| O15244 | 26 | 780 | Q15831 | 16 | 480 |
| O15245 | 27 | 810 | Q16820 | 21 | 630 |
| O35956 | 11 | 330 | Q5JUK3 | 19 | 570 |
| O60603 | 27 | 810 | Q5NUL3 | 139 | 4170 |
| O75751 | 16 | 480 | Q86UE8 | 10 | 300 |
| O95822 | 125 | 3750 | Q8NI60 | 11 | 330 |
| P08686 | 22 | 660 | Q8TDV5 | 354 | 10620 |
| P09884 | 16 | 480 | Q8TDX7 | 13 | 390 |
| P13807 | 25 | 750 | Q96RG2 | 20 | 600 |
| P16473 | 12 | 360 | Q96RJ0 | 438 | 13140 |
| P22888 | 10 | 300 | Q99808 | 34 | 1020 |
| P25021 | 182 | 5460 | Q9BRS2 | 17 | 510 |
| P25929 | 149 | 4470 | Q9GZN0 | 53 | 1590 |
| P28222 | 445 | 13350 | Q9H2X6 | 207 | 6210 |
| P28566 | 38 | 1140 | Q9H3N8 | 577 | 17310 |
| P30939 | 56 | 1680 | Q9HC97 | 191 | 5730 |
| P31213 | 124 | 3720 | Q9NXG6 | 33 | 990 |
| P32245 | 139 | 4170 | Q9RMS5 | 13 | 390 |
| P32249 | 35 | 1050 | Q9UHL4 | 202 | 6060 |
| P34969 | 942 | 28260 | Q9Y251 | 19 | 570 |
| P35414 | 40 | 1200 | Q9Y271 | 34 | 1020 |
| P35869 | 169 | 5070 | Q9Y5Y9 | 72 | 2160 |
| P40763 | 231 | 6930 | Q9Y616 | 31 | 930 |
| P43088 | 31 | 930 | Q9Y6L6 | 34 | 1020 |
| P43116 | 152 | 4560 | P47898 | 185 | 5550 |
| P49019 | 59 | 1770 | P56192 | 13 | 390 |