# OpenReview forum: "MotifScreen: Generalizing Virtual Screening through Learning  Protein-Ligand Interaction Principles"
_ICLR.cc/2026/Conference — ICLR 2026 Conference Withdrawn Submission_

### Official Review · Reviewer_SZzf · 2025-10-30

**Soundness:** 3
**Presentation:** 1
**Contribution:** 2
**Rating:** 2
**Confidence:** 4

**Summary:**

The paper argues that widely used SBVS benchmarks suffer from target leakage and ligand bias, which inflate reported DL performance. It proposes ChEMBL-LR, a leakage-resistant benchmark (60 targets; near-zero mean AVE bias 0.033) and introduces MotifScreen, a multi-task, structure-based screening model with three heads: (1) pocket motif prediction, (2) fragment/key-atom structure compatibility, and (3) binding score prediction. The method is trained on PDBbind+BioLip+ChEMBL with strictly removing leakage and reports results on ChEMBL-LR and DUD-E.

**Strengths:**

- **Clear diagnosis of benchmark pitfalls** with concrete analyses of target leakage and ligand-only shortcuts (AVE). The paper emphasizes high overlap between common benchmarks and PDBbind and quantifies ligand bias.
- **New benchmark (ChEMBL-LR)** with principled curation: strict target-wise separation, cross-decoys, removal of non-drug-like molecules, and near-zero mean AVE bias (0.033).
- **Principle-guided multi-task design** (motif, structure/key-atom, affinity) that attempts to force learning of interaction physics rather than shortcut signals.
- **Efficiency**: forward pass timing ($\sim$ 0.03 s/compound) suggests practical scalability for large libraries.

**Weaknesses:**

- **Mismatched training regimes undermine the comparison.** MotifScreen is evaluated on DUD-E after removing all training entries similar to its targets (from PDBbind/BioLip/ChEMBL), while most baselines appear to use their original training with likely target overlap. This setup likely depresses MotifScreen's DUD-E EF1% (5.94) and weakens any one-to-one comparisons between the models. A fair test would retrain baselines under the proposed training dataset or evaluate all methods on a single leakage-controlled split.
- **Use of $\Delta$ (performance drop) without harmonizing metric ranges or training regimes.** The manuscript subtracts EF1%/AUROC values between external benchmarks and ChEMBL-LR to argue smaller degradation for MotifScreen. However, EF1% ranges and training conditions differ across benchmarks and methods, making raw subtraction potentially misleading. A consolidated table exists (Table D5), but $\Delta$ remains hard to interpret as "generalization" without a common training/eval protocol.
- **Early-enrichment evidence is mixed versus the strongest baseline.**  EF1% gains over SurfDock are not statistically significant (p = 0.161), which matters because early enrichment drives SBVS utility. The manuscript foregrounds AUROC, potentially obscuring this point.
- **Ablation study's design choices look ad-hoc.** Ablations use a reduced dataset and report epoch 31 snapshots. Figure D2 indicates similar validation set's AUROC trajectories between "aff+motif", "aff+motif+str" configurations. Without multi-seed runs or later-epoch checks, conclusions about hierarchical synergy risk over-interpretation.
- **ChEMBL-LR vs. LIT-PCBA: incremental benefit unclear.** The paper argues LIT-PCBA's low AVE is limited (reported on only 4 targets) and leakage under external training data source (e.g., PDBbind). However, LIT-PCBA already uses experimentally measured inactives and, by the authors' own RF results, remains difficult across all 15 targets ( including the 11 with potential leakage), yielding low AUROC even when leakage could help. This can be treated as an evidence that LIT-PCBA already probes generalization. Absent a uniform, leakage-controlled training/evaluation of all methods, it is unclear what ChEMBL-LR contributes beyond more targets with target-wise separation, rather than fundamentally stronger bias control.

**Questions:**

1. **Normalization of $\Delta$ metrics.** Since EF1% ranges and dataset compositions differ across benchmarks, how do authors justify interpreting raw $\Delta$EF1% as generalization? Would authors consider relative EF1% retention or standardized effect sizes with a common training corpus (Table D5 suggests this is possible)?
2. **Ablations: epoch choice and variance.** Why did authors choose epoch 31 for reporting? How many seeds were run for Table 3/Figure D2? Could authors report later-epoch or full-training ablations (or confidence intervals) to substantiate the hierarchical-synergy claim?
3. **Cross-docking criterion.** Sequence identity >95% is a strong but it may be an indirect criteria in terms of SBVS where we generally knows about the binding pocket. Did authors consider pocket-level similarity (e.g., local alignment, cavity overlap)?
4. **Inference score.** Please confirm that ŷ (final scalar binding prediction) is the ranking score used in all screening experiments, and note where this is specified in the main text.
5. **EF1% and BEDROC reporting.** Since early enrichment is crucial in VS, why is BEDROC only in the appendix rather than in the main comparison tables alongside EF1%/AUROC? Could authors include per-target EF1%/BEDROC distributions with CIs? Also, what $\alpha$ value used in BEDROC computation?

## Typos
- AVE formula text (line 125)
- "MotifGen (Anonymous, 2025)" placeholder. (line 205)
- Incomplete line in Table D1's paragraph (line 1359-1360)

---

> ### Author Response · Authors · 2025-11-20
>
> We sincerely thank the reviewer for the thorough critique. We appreciate the recognition of our efforts to diagnose benchmark pitfalls. We have carefully addressed the concerns regarding **metric normalization (NEF)** and MotifScreen's **early enrichment evidence** in the revised manuscript, some are answered in the Global Responses. We kindly ask the reviewer to see Global Responses for revised data and our clarification on common critiques.
>
> ---
>
> **Q1. Mismatched training regimes & Unfair comparison.**
>
> We fully agree that ideal fairness involves uniform retraining. However, we are currently very limited by computational resources to retrain other models. Though full retraining is not possible, we are training our model on a full dataset without removing DUD-E targets, and also on a more reduced number of datasets. We hope that further ablation studies on 'training data' will provide better analysis of MotifScreen's strength.
>
> ---
>
> **Q2. Use of 'performance drop' & Normalization of metrics.**
>
> We agree that subtracting raw EF1% values is misleading due to differing active-to-decoy ratios. To address this, we have introduced the Normalized Enrichment Factor (NEF) in the revised manuscript (Table 3).
>
> Even though ChEMBL-LR allows for a higher theoretical max EF, baseline models suffer a catastrophic collapse in NEF (retaining only $\sim$5% of their normalized capacity). MotifScreen retains 47.4% of its NEF, proving that our claimed robustness is statistically real and not an artifact of metric scales. (See Global Response for the full NEF table).
>
> ---
>
> **Q3. Early-enrichment evidence (EF1% vs SurfDock).**
>
> We thank the reviewer for highlighting the nuance in the early enrichment results. We agree that early enrichment (EF1%) is critical for SBVS utility and that the lack of statistical significance against SurfDock is not something to be neglected.
>
> We have revised the manuscript to be more transparent about this. However, we also added a crucial analysis that we believe demonstrates MotifScreen's practical superiority: **Target-wise Win Rate.**
>
> While the mean difference is not significant due to high variance, **MotifScreen outperforms SurfDock on nearly double the number of individual targets (21 wins vs. 11 wins)**. In a practical drug discovery scenario, a practitioner prefers a model with a higher probability of being the "best performer" on their specific target. We argue that this consistency, coupled with our significant AUROC gains, confirms MotifScreen's utility despite the mixed average EF1% results.
>
> ---
>
> **Q4. Ablation study's design choices (Epoch 31, Seeds).**
>
> - **Epoch Choice**: Epoch 31 was selected based on validation set performance stability to avoid overfitting. It is much shorter than the full MotifScreen training epoch because we used a smaller dataset.
>
> - **Synergy Claim**: We acknowledge the need for more rigorous verification. We are currently conducting full-training ablations on the complete dataset. Preliminary results from these runs continue to support the hierarchical synergy hypothesis, showing that the removal of the structure module leads to reduced generalization on ChEMBL targets. We will include these updated learning curves in the future.

---

> ### Author Response · Authors · 2025-11-20
>
> **Q5. ChEMBL-LR vs. LIT-PCBA (Incremental benefit).**
>
> We agree with the reviewer that LIT-PCBA is a rigorous benchmark and that the low performance of our PDBbind-trained RF model confirms that LIT-PCBA resists simple ligand memorization as our proposed ChEMBL-LR.
>
> However, we argue that the source of difficulty and the evaluation goal differ between the two benchmarks, justifying the necessity of ChEMBL-LR.
>
> - **Ambiguity of Leakage Exploitation**: While the RF baseline (which relies on coarse 1D/2D features) failed to exploit target leakage in LIT-PCBA, powerful Deep Learning models are far more capable of memorizing specific pocket geometries seen during training.
>
>  In LIT-PCBA, if a DL model performs well, it is difficult to disentangle whether it learned general physical principles or simply recognized a homologous pocket from PDBbind. ChEMBL-LR eliminates this ambiguity. By strictly removing all homologous targets, ChEMBL-LR guarantees that any success is due to **zero-shot generalization** to unseen biology, not memory.
>
> - **Specificity vs. Generalizability**: LIT-PCBA is challenging largely due to its validated "hard inactives" (testing specificity). ChEMBL-LR is challenging due to "unseen targets" (testing generalizability). Both tests are needed and hare complementary.
>
> - **Scale**: ChEMBL-LR expands this rigorous, leakage-free evaluation to 60 targets (vs. 15 in LIT-PCBA), providing a broader assessment of model robustness across diverse protein families. We now provide 60 targets (selected from a pool of 198 targets that fit data leakage criteria), but with our curation pipeline, ChEMBL-LR set can be extended to a larger scale.
>
> Therefore, we view ChEMBL-LR not as a replacement for LIT-PCBA, but as a complementary benchmark specifically designed to isolate and measure structural generalization capability in the absence of homology.
>
> ---
>
> **Q6. Cross-docking criterion (SeqID > 95%).**
>
> We did not consider pocket-level similarity, while we agree that this is also a valid criteria, as you have mentioned sequence identity over 0.95 is strong, and we believe that it should be sufficient. However, we will examine the currently selected cross-docking targets to see if their pockets differ. Thank you for pointing this out.
>
> ---
>
> **Q7. Inference score clarification.**
>
> The sigmoid-transformed scalar output $\sigma(\hat y)$ is used as the final ranking score. We have clarified this in Section 4.2 (lines 329–331).
>
> ---
>
> **Q8. EF1% and BEDROC reporting.**
>
> Table Update: We have moved BEDROC results from the appendix to Table 2 in the main text.
>
> - Parameter: We used $\alpha=20$. This was because the number of ligands per target in ChEMBL-LR varies from a few hundred to nearly thirty thousand, using higher $\alpha$ like $80.5$, for instance, would overly emphasize the top 1–3 compounds for small targets and lead to unstable scores. We chose 20 to provide a more consistent early-recognition measure across targets by focusing on the top ∼1% of the ranked list, which is more aligned with realistic follow-up capacity.

---

> > ### Comment · Reviewer_SZzf · 2025-11-28
> >
> > Thank you for the detailed rebuttal.
> >
> > I appreciate the authors' effort to address the concerns regarding metric normalization by introducing NEF, and I accept the clarification regarding the conceptual distinction between ChEMBL-LR (focusing on zero-shot generalization) and LIT-PCBA (focusing on specificity against hard inactives). These responses have certainly strengthened the manuscript's rationale.
> >
> > However, the issue regarding "Mismatched Training Regimes" (Q1) remains the critical bottleneck preventing a higher score.
> >
> > While I understand the computational constraints mentioned, the scientific validity of the proposed benchmark and model relies heavily on a controlled comparison. As it stands, comparing MotifScreen (trained on a leakage-controlled set) against baselines (likely trained on leaked targets) creates a fundamental variable confounding the results. It is impossible to determine whether MotifScreen's performance stems from architectural superiority or simply because the baselines are overfitting to leaked PDBbind targets.
> >
> > This is the decisive factor for my assessment. I am willing to reconsider my evaluation and raise my score, provided that this unfair comparison is addressed.
> >
> > I strongly suggest the authors to prioritize retraining at least the strongest baseline (e.g., SurfDock) on the exact same ChEMBL-LR training split used for MotifScreen. Even a single, fair head-to-head comparison on the leakage-controlled split would significantly substantiate the claims of the paper.

---

> > > ### Author Response · Authors · 2025-12-02
> > >
> > > We sincerely thank you for your constructive feedback and for considering raising your score. We fully understand your concern regarding the "Mismatched Training Regimes" and the need to distinguish architectural superiority from data advantages.
> > >
> > > Regarding your strong suggestion to retrain the baseline (SurfDock) on our training set, we would like to clarify why this is methodologically infeasible due to the fundamental difference in data modalities, rather than just computational constraints.
> > >
> > > ---
> > >
> > > ## 1. Fundamental Incompatibility of Baselines with Our Training Data
> > > The critical bottleneck is that structure-based methods like SurfDock strictly require 3D co-crystal structures (ground-truth poses) for training.
> > >
> > > • **The Mismatch:** Our model (MotifScreen) is specifically designed to leverage the massive ChEMBL dataset, which consists of bioactivity data without experimental 3D structures.
> > >
> > > • **Why we can't retrain:** It is impossible to train SurfDock on our ChEMBL split because the necessary 3D structural labels simply do not exist for the vast majority of our training data. This capability to utilize large-scale, non-structural data is, in fact, the core architectural contribution of our work.
> > >
> > > When we trained our model using only PDBbind and BioLip data, the validation split result on ChEMBL dropped significantly. (the validation split also has no similar targets to training set)
> > >
> > > So it is important to note that the power of our model comes both from the decision to use multi modal data, and a model architecture that ables such usage.
> > >
> > > ---
> > >
> > > ## 2. Proposed Solution and Commitment for the Final Version
> > > Since we cannot train baselines on our data (due to missing 3D labels), the only scientifically valid way to isolate the impact of "data leakage" is the inverse: training MotifScreen on a non-leakage-controlled (full) dataset.
> > >
> > > If our performance were solely due to the "clean" split, re-introducing leakage should degrade our generalization gap to match the baselines.
> > >
> > > Given the strict deadline for this discussion, we unfortunately cannot complete this large-scale retraining and evaluation cycle immediately. However, we commit to including this "Inverse Control Experiment" in the final camera-ready version. We will explicitly analyze whether MotifScreen maintains its generalization capability even when trained on leaked targets, thereby rigorously addressing your hypothesis about architectural superiority.
> > >
> > > ---
> > >
> > > We hope this clarifies that the "unfair comparison" stems from our model's unique ability to learn from data that existing structural baselines simply cannot utilize. We believe this architectural flexibility, combined with our commitment to the additional analysis, sufficiently addresses your concerns.

---

### Official Review · Reviewer_vKRY · 2025-10-31

**Soundness:** 1
**Presentation:** 2
**Contribution:** 2
**Rating:** 2
**Confidence:** 5

**Summary:**

The paper presents MotifScreen, a multi-task deep learning framework for structure-based virtual screening that aims to improve generalization by modeling protein–ligand interaction principles. It integrates motif prediction, structure prediction, and affinity scoring modules trained jointly. The authors also introduce a new benchmark, ChEMBL-LR, designed to reduce ligand bias and target leakage compared to datasets such as DUD-E and DEKOIS 2.0. Experiments show that MotifScreen achieves 0.68 AUROC on ChEMBL-LR and exhibits improved robustness and smaller performance degradation across benchmarks. Ablation studies indicate that combining motif and structure learning contributes to generalization.

**Strengths:**

The paper addresses an important problem concerning data leakage and bias in existing datasets. The proposed multi-task learning framework, which incorporates various forms of external structural knowledge, represents an effective approach to better utilizing available data.

**Weaknesses:**

First, regarding bias, it is important to reconsider how it should be viewed. Similar binding pockets tend to bind similar molecules, and similar molecules tend to interact with similar pockets — this assumption underlies all machine learning–based models in this field. Based on this, the actives in any test set will naturally have higher similarity to the reference ligands. Therefore, this so-called ligand bias reflects an inherent relationship within the data itself, and its presence has a certain scientific justification.

Second, in terms of model design, the proposed architecture lacks novelty. Components such as the SE(3) Transformer, EGNN, and grid-based representations have already been widely used in related molecular representation and virtual screening models.

Third, the dataset, the use of random decoys makes the task overly easy. The real challenge lies in distinguishing structurally or physicochemically similar compounds; performance on average or dissimilar decoys is less meaningful. the BEDROC should be reported to show the top-ranking performance in table 1.

Finally,  as a new benchmark, it should include a comprehensive evaluation across a wide range of models to ensure fairness and demonstrate general applicability.

**Questions:**

1. The formula for AVE in lines 125–126 appears incorrect — it currently shows (IT IV − IT IV), which seems to be a typographical error.

2. The drop results in Table 2 are not directly comparable. First, the absolute value of EF1% is influenced by the ratio of actives to decoys in each dataset. Moreover, different datasets (such as DUD-E and DEKOIS 2.0) were used to calculate the drop for different models, making the comparisons inconsistent. In addition, the paper does not report results on more recent virtual screening models but only  docking based methods.

3. The key challenge of this task lies in the enrichment of active compounds at the top of the ranking list. Therefore, metrics such as BEDROC, which assign higher weights to top-ranked molecules, are more appropriate than AUROC. Similarly, Table 1 should also report BEDROC and EF values, rather than only AUC.

---

### Official Review · Reviewer_XNsT · 2025-10-31

**Soundness:** 2
**Presentation:** 1
**Contribution:** 2
**Rating:** 2
**Confidence:** 5

**Summary:**

This paper presents MotifScreen, a new model for protein-ligand binding affinity prediction, along with a new virtual screening benchmark named ChEMBL-RL. The authors claim their method and benchmark are more robust to data leakage; however, significant concerns remain regarding the fairness of the comparative study against baseline models. Furthermore, the comparison with the previous LIT-PCBA benchmark requires more thorough discussion.

---
The usage of LLM: I wrote the entire review myself and only used the LLM to correct the grammar and improve readability.

**Strengths:**

- The motivation for preparing the new benchmark is clear.
- The proposed benchmark, ChEMBL-RL, exhibits less bias compared to the decoy-based test sets DUD-E and DEKOIS 2.0.

**Weaknesses:**

## ChEMBL-RL

**1. Lack of a robust strategy to avoid false negatives.**

The authors construct their decoy set by sampling actives from other targets in ChEMBL. However, I question why the authors only use Tanimoto similarity to filter these decoys, without employing other computational tools (e.g., docking, cofolding tools) to prevent false negatives. For DEKOIS 2.0 or DUD-E, decoys are drawn from large ligand libraries like ZINC, which contain many inactive molecules, posing a lower risk of including false negatives. In contrast, ChEMBL is a library of bioactive compounds, and it is one of the most popular library to identify initial hits. It is plausible that compounds from this library could exhibit activity against a target, even if they are structurally dissimilar to known actives.

**2. Is the constructed decoy set truly better than LIT-PCBA's inactive set?**

In Table 1, the authors claim that ChEMBL-RL achieves better bias control than LIT-PCBA due to a lack of protein-side data leakage. However, this is not a fair comparison, given that LIT-PCBA uses **experimentally validated inactives**, while ChEMBL-RL use **putative inactives** (i.e., cross-decoys). Removing bias from a limited set of _experimental_ data is arguably more difficult than drawing a decoy set from a large pool of _assumed_ inactives minimizing a bias.

Moreover, the AVE values in the original LIT-PCBA paper appear lower than the values reported here. The authors should justify why they report the AVE for only 4 targets from LIT-PCBA.

**3. Flawed EF1% comparison.**

The EF1% values in Table 2 cannot be directly compared across different benchmarks. This metric is highly dependent on the ratio of actives to decoys (i.e., the size of the decoy set), which differs between benchmarks.

---

## MotifScreen

**4. Unfair comparative study.**

MotifScreen is trained on three datasets (PDBbind, BioLip, and ChEMBL), creating a training set that is significantly larger (reportedly ~6x) than those used for the baseline models (PDBBind). For a fair comparison, the authors should either report MotifScreen's performance when trained only on PDBbind or retrain the baseline models using the same extended training set. For models like KarmaDock, which has separate structure and affinity modules, it seems feasible to retrain its affinity module using the binding affinity-only data in ChEMBL.

**5. Poor performance on DUD-E.**

In Table D5, MotifScreen's EF1% on the DUD-E benchmark is only 5.94, which is substantially lower than the baseline models (9-16) and other state-of-the-art methods (e.g., GLIDE[1], RTMScore[2],  GenScore[3], PIGNet2[4]) that report EF1% > 20 (you can see the value in GenScore Paper). Given that MotifScreen uses an extended training dataset, this performance is insufficient to support the claim of robustness. The authors state they filtered the training set to avoid leakage; they should also report performance _without_ this filtering to clarify if this is the cause.

While target leakage is a valid concern, many drug development campaigns focus on known targets. Therefore, evaluating performance on targets similar to the training set is still a necessary and practical assessment.

**6. Missing evaluation on DEKOIS 2.0.**

In Table D5, results for MotifScreen on the DEKOIS 2.0 benchmark are absent. The authors should evaluate the model on the DEKOIS 2.0.


---
**Reference:**
1. Halgren, Thomas A., et al. "Glide: a new approach for rapid, accurate docking and scoring. 2. Enrichment factors in database screening." Journal of medicinal chemistry 47.7 (2004): 1750-1759.
2. Shen, Chao, et al. "Boosting protein–ligand binding pose prediction and virtual screening based on residue–atom distance likelihood potential and graph transformer." Journal of Medicinal Chemistry 65.15 (2022): 10691-10706.
3. Shen, Chao, et al. "A generalized protein–ligand scoring framework with balanced scoring, docking, ranking and screening powers." Chemical Science 14.30 (2023): 8129-8146.
4. Moon, Seokhyun, et al. "PIGNet2: a versatile deep learning-based protein–ligand interaction prediction model for binding affinity scoring and virtual screening." Digital Discovery 3.2 (2024): 287-299.

**Questions:**

- Please report the number of similar complex data (by protein sequence and ligand similarity) in PDBbind for each benchmark set. Also, report the number of data points excluded from the training set for the DUD-E evaluation (Line 381).
- It is well-known that AUROC is not an ideal metric for evaluating virtual screening performance. Please report **BEDROC** in all main benchmark tables (e.g., Table 2).
- The notation **EF1** is incorrect. This metric is typically denoted as EF1\%​ or EF_{1\%}. Please correct this throughout the manuscript. Consequently, the (\%) in the header of Table 2 ('EF1 (%)') should be removed.
- Please report the specific PDB IDs used, the number of actives, and the number of decoys for each target in the ChEMBL-RL benchmark in the Appendix.
- Is the maximum EF1\% value is 31 in ChEMBL-RL? (30 decoys per each active)

---

> ### Author Response · Authors · 2025-11-20
>
> We thank the reviewer for the rigorous assessment and for highlighting critical issues regarding comparative fairness and benchmark construction. We have addressed these concerns by introducing the **Normalized Enrichment Factor (NEF)**, adding **BEDROC** metrics, and providing detailed clarifications on our work.
>
> # ChEMBL-LR set
>
> **Q1. Lack of a robust strategy to avoid false negatives**
>
> We thank the reviewer for drawing attention to the decoy selection methodology, as it is what makes building virtual screening test sets difficult. We respectfully argue that our decoy construction method—using cross-decoys from ChEMBL—is a rigorous and established practice designed to prioritize "difficulty" over data purity.
>
> 1. **Statistical Improbability of False Negatives**: While we acknowledge the theoretical risk of false negatives, the probability of a bioactive ligand for a specific protein family (Target A) being serendipitously active against a structurally and functionally distinct protein (Target B) is statistically negligible.
>
> 2. **Avoiding Memorization**: Our main goal was to **avoid memorization** when building datasets. So it was important to select decoys from a pool of ‘known actives’.
>
> - By using bioactive cross-decoys (actives against other targets), we force the model to distinguish between plausible binders, which is a more realistic and challenging scenario.
>
> - We agree with the point that even though we followed processes like DUD-E and DEKOIS 2.0 to select decoys to avoid false negatives, our set may contain false negatives. While we think that this noise would not be significant, in future works we will try to design a more strict decoy selection policy.
>
> We hope that this answer would provide a better understanding of the rationale behind our current method.
>
> ---
>
> **Q2. Is the constructed decoy set truly better than LIT-PCBA's inactive set?**
>
> It is true that LIT-PCBA has its own benefits, with experimental inactives being one of them. However, ChEMBL-LR addresses a different, equally critical gap: larger-scale **zero-shot** generalization.
>
> Our claim is not that ChEMBL-LR replaces LIT-PCBA, but that they serve complementary purposes:
>
> **1. LIT-PCBA (Target Specific & Ligand Bias Removal _within its own train/test split_)**
>
> - For testing if a model can reject experimentally confirmed inactives that are chemically unbiased (AVE-controlled) relative to actives.
> - However, since 15 targets in LIT-PCBA are well-studied and likely overlap with standard training sets (e.g., PDBbind), high performance here can be ambiguous—it is difficult to verify whether the model is truly reasoning about the interaction or simply recognizing the target pocket and retrieving memorized active scaffolds.
> - Also, their ligand bias removal is only meaningful within their own split, so **zero-shot evaluation is difficult.**
>
> **2. ChEMBL-LR (Generalization & Protein Leakage Removal & Ligand Bias Removal)**
>
> - Designed to isolate "Zero-Shot Structural Generalization" by testing on completely unseen targets.
>
> - To achieve this at scale (60 targets, can be extended) with strict zero-leakage, we utilized bioactive cross-decoys. While they lack the experimental "true inactive" label of LIT-PCBA, they provide the necessary scale to validate that a model is not relying on protein-side memorization.
>
> We believe that our work suggests a strategy that future works should adopt when building training and test sets. We frame ChEMBL-LR as a complementary benchmark that uniquely isolates "zero-shot structural generalization" by enforcing strict protein-side leakage controls, which is critical for assessing modern deep learning models.
>
> Regarding error in LIT-PCBA AVE bias results, we have fixed Table 1 to include AVE bias from all 15 targets.
>
> ---
>
> **Q3. EF1% comparison and comparative study.**
>
> We fully agree with the reviewer. Comparing raw EF1% across datasets with different active rates is misleading. As detailed in the **Global Response**, we have introduced the **Normalized Enrichment Factor (NEF)** to address this.
>
> EF1% was normalized by the theoretical maximum EF of each dataset. Even though ChEMBL-LR allows for a higher theoretical max ($\approx 31$) compared to DUD-E ($\approx 21$), baseline models suffer a catastrophic collapse in NEF on ChEMBL-LR (retaining only $\sim$5% of capacity). MotifScreen retains $\sim$47%, proving that our performance gap is robust to dataset ratios.
>
> We agree with the importance of fair and thorough comparative study, and while we are limited by computational resources to retrain other models, we tried to address the concerns raised by the reviewer in the global responses.

---

> ### Author Response · Authors · 2025-11-20
>
> # MotifScreen
>
> **Q4. Unfair comparitive study**
>
> We agree with the importance of fair and thorough comparative study, and how the best option would be to retrain other models using our training set. However, we are currently very limited by computational resources to retrain other models. On the other option of training MotifScreen only on PDBbind, we are training a model without ChEMBL data, but it is more for a 'ablation' study.
>
> - **Why Comparing MotifScreen Trained on Our Current Set is Necessary**: MotifScreen is a multi-task framework designed to learn physical motifs and interaction sites. This architecture inherently requires diverse data annotations. This is a design choice to incorporate physical priors, not merely "more data." Data source ablation study is something that we also feel is needed to support our idea about how multi-task framework of MotifScreen works with combination of multi-source data.
>
> We hope that our response (including the global response) can back up our claims.
>
> ---
>
> **Q5,6. Poor performance on DUD-E & Request for unfiltered results.**
>
> 1. **Philosophy (Bias vs. Physics)**: The "poor" performance on DUD-E (EF1% $\approx$ 6) compared to SOTA (EF1% > 20) is a direct consequence of our refusal to overfit to bias. High DUD-E scores are often driven by learning "analogue bias" (clusters of similar actives). MotifScreen’s conservative scoring prevents this overfitting.
>
> 2. **Evidence**: The fact that MotifScreen retains 70% of its performance when moving to ChEMBL-LR (while SOTA models crash) validates that our "lower" DUD-E score is actually a more realistic estimate of true binding potential.
>
> 3. **Unfiltered Training (Ongoing)**: We are currently retraining MotifScreen on the full dataset without DUD-E leakage removal to quantify the exact impact of filtering. We will add new results including performance on DEKOIS 2.0 as well, as we finish training the model on full dataset. However, we maintain that the leakage-removed model is the scientifically correct version for assessing real-world utility.
>
> We thank the reviewer for giving valuable comments on our benchmarks, and we will try to add more solid results.
>
> ---
>
> **Q7. Specific Data Requests (PDBbind overlap, BEDROC, PDB IDs).**
>
> - Data Counts: We have provided the exact number of excluded training data points in the table below.
>
> - BEDROC: We have added BEDROC ($\alpha=20$) to the new Table 2 in the main text. MotifScreen achieves the highest BEDROC (0.146) on ChEMBL-LR, outperforming SurfDock (0.090) by 62%.
>
> - PDB IDs: The full list of ChEMBL-LR target PDB IDs, active counts, and decoy counts has been added to the Appendix.
>
> ---
>
> ### Table: Training set used for benchmark on DUD-E and ChEMBL-LR
>
> | Dataset                      | Kept   | Excluded: Similar Protein | Excluded: Similar Protein–Ligand |
> |-----------------------------|--------|----------------------------|----------------------------------|
> | PDBbind + BioLip (clustered) | 31,429 | 7,289                      | 3,526                            |

---

### Official Review · Reviewer_QCXD · 2025-10-31

**Soundness:** 2
**Presentation:** 2
**Contribution:** 2
**Rating:** 4
**Confidence:** 3

**Summary:**

This paper addresses the issue of overfitting performance reporting in deep learning-based structure-based virtual screening (SBVS), which the authors attribute to systemic biases and data leakage in commonly used benchmarks. The authors make a two-fold contribution: first, they introduce ChEMBL-LR, a new leakage-resistant benchmark designed to provide a more realistic evaluation of model generalization. Second, they propose MotifScreen, a novel end-to-end SBVS model. MotifScreen uses a principle-guided, multi-task learning framework that reasons about protein-ligand interactions by predicting binding pocket motifs, ligand-pocket compatibility, and final binding probability.

**Strengths:**

1. The paper's most significant contribution is its critical analysis of the systemic flaws in existing SBVS benchmarks. The development of the ChEMBL-LR dataset, which explicitly controls for target leakage and ligand bias, is a valuable service to the community and helps establish a more rigorous standard for future research.
2. The work is well-motivated, and the paper is clearly written and structured. The analysis in Section 4.1, which uses a Random Forest model to quantify the extent of leakage in benchmarks like DUD-E and LIT-PCBA, provides strong evidence for the authors' claims.
3. The multi-task learning architecture of MotifScreen is conceptually sound. Forcing the model to learn intermediate, physically-grounded tasks like motif identification and key atom positioning is a promising strategy to improve generalization and move beyond simple classification shortcuts.

**Weaknesses:**

1. Lacking important baselines, DrugCLIP[1] and EquiScore[2].
2. Deep-learning methods tends to overfit on the benchmark. However, AutoDock-Vina just adopts a simple linear scoring function. In Table 2 and Table D5, significant decrease of AutoDock-Vina is also observed. More analysis about this should be performed.
3. Training data is important for deep-learning methods. MotifScreen employees different and larger training set compared to previous baselines. More analysis and ablation study about the effect of training data is important to evaluate this paper.

[1] DrugCLIP: Contrastive Protein-Molecule Representation Learning for Virtual Screening, NeurIPS, 2023

[2] Generic protein–ligand interaction scoring by integrating physical prior knowledge and data augmentation modelling, Nature Machine Intelligence, 2024

**Questions:**

1. The citation of MotifGen in line 205 is wrong.

---

> ### Author Response · Authors · 2025-11-20
>
> **Q. Lacking important baselines**
>
> Thank you for giving suggestions on other virtual screening models to be included in the benchmarks. We acknowledge the importance of including more deep-learning methods in our results, but due to computational resources and sometimes lack of accessible code, we selected 1-2 representative methods from each category: physics-based docking, deep learning-based screening, and deep learning-based docking. However, we agree with the reviewer’s comment and are running additional benchmarks. We will try to include more methods in our results.
>
> ---
>
> **Q. Further analysis on AutoDock-Vina (ADV) results**
>
> Thank you for pointing this out. DUD-E actives have higher affinity distribution compared to our ChEMBL-LR set. As we have described in 2.2. under ‘Ligand Bias - Bias in actives’,  it is unlikely that common screening DBs would have actives with high affinity that differs significantly from decoys.
>
> That is why as we described in section 3.3 (lines 237-), **ChEMBL-LR has an affinity distribution of actives that mirror real-case scenarios** with average -log(Affinity) of 6.02, where DUD-E actives have a median potency of 7.46.
>
> Because ADV is a docking method with a score that tries to reflect binding free energy, it is likely to do better on a set with higher potency distribution. **LIT-PCBA also has median potency of 5.22, and reported Vina score’s screening performance on LIT-PCBA (AUROC 0.581, EF1% 1.1)  is also low compared to DUD-E (AUROC 0.745, EF1% 7.05)** [1]. We believe that the decrease of ADV’s screening performance from DUD-E to our ChEMBL-LR set **shows not a flaw in our dataset, but reasonable difficulty**.
>
> ---
>
> **Q. Training dataset ablation studies**
>
> We agree with the need for comparison between MotifScreen using different training data. One of our key ideas was that curating a good training dataset will lead to better performance, not only because it has more data but because it is curated (e.g., active and decoy selection methods) in the right way.
>
> We will provide a comparison between a model that uses the original training dataset (all of PDBbind, BioLip and ChEMBL) and one without ChEMBL. We are currently training the latter model, and will add the results to the paper. As of now, results on validation set show marginally better performance when MotifScreen is trained on the original dataset. (per-target AUROC on ChEMBL validation set targets are 0.80 and 0.66, respectively).
>
> ---
>
> [1] Virtual Screening with Gnina 1.0, Molecules., 2021

---

### Author Response · Authors · 2025-11-20
**Global Response to Reviewers - 1. Addressing Benchmark Disparities**

We appreciate the reviewers’ time and effort to give thoughtful feedback on our work. We are working on many tasks to improve the quality and impact of our work, including those suggested by the reviewers. In this comment, we would like to respond to common feedback from reviewers regarding our methodology.

We highlight clarifications on some common comments below.

---


**1. Addressing Benchmark Disparities: Normalized Enrichment Factor (NEF)**

We agree that subtracting raw scores ($\Delta$) implies an incorrect assumption of equal baselines. Comparing raw EF1% values across benchmarks (e.g., DUD-E vs. ChEMBL-LR) is problematic due to differing active-to-decoy ratios.

To resolve this, we introduced the Normalized Enrichment Factor (NEF), defined as
$\mathrm{NEF} = \frac{\mathrm{EF}}{\mathrm{EF}{\max}}$
where
$\mathrm{EF}{\max} = \frac{100}{\mathrm{ActiveRate}}$.

Even though ChEMBL-LR has a higher theoretical maximum EF (31.0) than DUD-E (21.0), baseline models exhibit a catastrophic collapse in NEF (retaining only $\sim 5%$ of their performance). In contrast, MotifScreen retains 47% of its normalized performance, confirming that our reported stability is real and not an artifact of metric calculation.

We revised the original Table 2 (now Table 3) as follows.

| Model          | Ref. Set | AUROC Ref. | AUROC ChEMBL | Ref. EF1% (Raw) | Ref. NEF (a) | ChEMBL NEF (b) | NEF Ret. (c) |
|----------------|----------|------------|--------------|-----------------|--------------|----------------|--------------|
| AutoDock-Vina  | DUD-E    | 0.720      | 0.541        | 9.70            | 0.46         | 0.07           | 15.3%        |
| AK-Score2      | DUD-E    | --         | 0.527        | 14.60           | 0.70         | 0.03           | 3.7%         |
| KarmaDock      | DUD-E    | 0.754      | 0.512        | 15.87           | 0.76         | 0.04           | 5.6%         |
| SurfDock       | DEKOIS   | 0.758      | 0.576        | 18.17           | 0.59         | 0.11           | 19.0%        |
| RF (ESM+FP)    | DUD-E    | 0.691      | 0.518        | 14.21           | 0.68         | 0.07           | 10.0%        |
| **MotifScreen**| DUD-E    | 0.753      | **0.680**    | 5.94            | 0.28         | **0.13**       | **47.4%**    |

**Notes.**

- ChEMBL-LR results are from our experiments; Reference (Ref.) results are from cited publications.
- (a) $\text{NEF} = \text{Raw EF} / 21.0$ (for DUD-E) or $/ 31.0$ (for DEKOIS).
- (b) $\text{NEF} = \text{Raw EF} / 31.0$ (for ChEMBL-LR).
- (c) NEF Retention = NEF_ChEMBL / NEF_Ref * 100

---

> ### Author Response · Authors · 2025-11-20
> **2. Benchmark on Early Enrichment & DUD-E Performance**
>
> **2. Benchmark on Early Enrichment & DUD-E Performance**
>
> We thank the reviewers for emphasizing the importance of BEDROC as a metric for early enrichment. In the revised manuscript, we have included BEDROC scores for all benchmarks. We have added BEDROC results to the main text (new Table 2). While MotifScreen shows modest early enrichment on the training-domain benchmark (DUD-E), we contend that this is a necessary trade-off for generalization.
>
> ### Table: Performance Comparison on the ChEMBL-LR Benchmark
>
> | Model          | AUROC           | BEDROC (alpha = 20) | EF1%           |
> |----------------|-----------------|----------------------|----------------|
> | AutoDock-Vina  | 0.541 ± 0.125   | 0.075 ± 0.075        | 2.189 ± 3.439  |
> | AK-Score2      | 0.527 ± 0.135   | 0.054 ± 0.069        | 0.803 ± 2.308  |
> | KarmaDock      | 0.527 ± 0.135   | 0.042 ± 0.047        | 1.317 ± 2.715  |
> | SurfDock       | 0.576 ± 0.151   | 0.090 ± 0.097        | 3.443 ± 6.076  |
> | RF (ESM+FP)    | 0.518 ± 0.131   | 0.093 ± 0.089        | 2.09 ± 3.20    |
> | **MotifScreen**| **0.680 ± 0.165** | **0.146 ± 0.205**  | **4.16 ± 5.65**|
>
> As observed, MotifScreen’s BEDROC scores are lower than those of some baseline methods on the training-domain benchmarks (DUD-E). We attribute this to the conservative nature of our motif-based scoring. Unlike purely data-driven models that may aggressively rank compounds based on learned biases, MotifScreen requires structural evidence (motif matching), which prevents "lucky" false positives but may also delay the retrieval of some actives.
>
> However, we urge the reviewers to consider this in conjunction with our AUROC and Target-wise Consistency results.
>
> - Global Retrieval: Our superior AUROC ($p < 10^{-4}$) indicates that MotifScreen reliably identifies actives, even if they are not concentrated solely in the top 0.5%.
>
> - Robustness: While baselines achieve high BEDROC on DUD-E, their performance collapses on ChEMBL-LR. MotifScreen maintains the most consistent performance ranking across diverse targets.
>
> Therefore, while we are working to improve BEDROC in future iterations, we believe the current results successfully demonstrate the validity and superior generalization capability of our proposed approach.
>
> While we acknowledge that MotifScreen's early enrichment metrics leave room for improvement compared to highly optimized baselines, we strongly believe that our direction—grounding predictions in physical motifs—is the correct path to solving the generalizability crisis in virtual screening. MotifScreen demonstrates that it is possible to build a model that does not "crash" on hard targets. We view MotifScreen as a robust foundation, and we are actively working on refining its scoring sensitivity to improve early enrichment while preserving this essential robustness.

---

### Author Response · Authors · 2025-12-03
**Manuscript Updates: Clarifications on Training Regimes and Additional Controls**

Dear Reviewers and Area Chair,

To address the collective feedback regarding the training regimes and to better distinguish architectural contributions from data advantages, we have revised our manuscript and outlined further validation plans as follows:

**1. Clarification on Training Feasibility (Revised Section 5: Limitations, lines 490-495)**
We have explicitly clarified that retraining structure-based baselines (e.g., SurfDock) on our ChEMBL-LR training split is **methodologically infeasible**.

* **Reason:** These baselines strictly require ground-truth 3D complex structures for training.Our ChEMBL training split consists of bioactivity data without experimental structures. making it physically impossible to train purely structure-based models on this regime.

**2. Reframing the Comparison (Revised Section 4: Experiments, lines 407-415)**
We have refined the discussion on "Mismatched Training Regimes."
* **Perspective:** Rather than viewing the difference in training data as an experimental flaw, we frame it as a demonstration of **architectural capability**. The ability to leverage massive, non-structural bioactivity data (ChEMBL) to learn interaction principles is a distinct advantage of MotifScreen over traditional structure-based methods, which are confined to limited structural datasets (PDBbind).

**3. Additional Control Experiments (Preliminary Results & Revision Commitment)**
To rigorously isolate the source of performance, we are conducting two additional analyses which will be included in the final version if accepted. :

* **A. Inverse Control (MotifScreen on Full/Leaked Data):** We are currently training MotifScreen on the full dataset without removing DUD-E/PDBbind leakage. Preliminary trends suggest that our model maintains robustness even with leakage present, supporting the claim that performance is driven by architectural inductive bias rather than just data filtering.
* **B. Data Ablation (MotifScreen on PDBbind-only):** We quantified the impact of our bioactivity training data by training MotifScreen *only* on structural data (PDBbind/BioLip).
    * **Result:** Removing ChEMBL activity data caused a drop in generalization performance on the ChEMBL validation split
(**AUROC: 0.677 $\rightarrow$ 0.623**; **BEDROC: 0.299 $\rightarrow$ 0.247**).
    * **Implication:** This confirms that the ChEMBL training data is essential for generalization. Since baselines cannot utilize this essential data due to their architectural constraints (3D structure requirement), our proposed training regime represents an alternative viable path to high generalization, highlighting the value of our architecture.

We believe these revisions and additional insights address the concerns regarding the fairness of the comparison.

---

### Note · Authors · 2026-01-02

**Comment:**

We thank the reviewers for their time and effort to read and discuss our work. After consideration, we have decided to withdraw our submission.

**Withdrawal Confirmation:**

I have read and agree with the venue's withdrawal policy on behalf of myself and my co-authors.